# An Ensemble of Deep Learning Object Detection Models for Anatomical and Pathological Regions in Brain MRI

**DOI:** 10.3390/diagnostics13081494

**Published:** 2023-04-20

**Authors:** Ramazan Terzi

**Affiliations:** 1Department of Big Data and Artificial Intelligence, Digital Transformation Office of the Presidency of Republic of Türkiye, Ankara 06100, Turkey; ramazan.terzi@cbddo.gov.tr or ramazan.terzi@amasya.edu.tr; 2Department of Computer Engineering, Amasya University, Amasya 05100, Turkey

**Keywords:** anatomical and pathological object detection, model ensemble, benchmark, brain MRI

## Abstract

This paper proposes ensemble strategies for the deep learning object detection models carried out by combining the variants of a model and different models to enhance the anatomical and pathological object detection performance in brain MRI. In this study, with the help of the novel Gazi Brains 2020 dataset, five different anatomical parts and one pathological part that can be observed in brain MRI were identified, such as the region of interest, eye, optic nerves, lateral ventricles, third ventricle, and a whole tumor. Firstly, comprehensive benchmarking of the nine state-of-the-art object detection models was carried out to determine the capabilities of the models in detecting the anatomical and pathological parts. Then, four different ensemble strategies for nine object detectors were applied to boost the detection performance using the bounding box fusion technique. The ensemble of individual model variants increased the anatomical and pathological object detection performance by up to 10% in terms of the mean average precision (mAP). In addition, considering the class-based average precision (AP) value of the anatomical parts, an up to 18% AP improvement was achieved. Similarly, the ensemble strategy of the best different models outperformed the best individual model by 3.3% mAP. Additionally, while an up to 7% better FAUC, which is the area under the TPR vs. FPPI curve, was achieved on the Gazi Brains 2020 dataset, a 2% better FAUC score was obtained on the BraTS 2020 dataset. The proposed ensemble strategies were found to be much more efficient in finding the anatomical and pathological parts with a small number of anatomic objects, such as the optic nerve and third ventricle, and producing higher TPR values, especially at low FPPI values, compared to the best individual methods.

## 1. Introduction

Imaging technologies are frequently used for disease detection and evaluation in the medical domain to overcome the uncertainty and limited detectability of human perception. Different imaging methods such as magnetic resonance imaging (MRI), positron emission tomography (PET), single-photon emission computed tomography (SPECT), ultrasound, computed tomography (CT), and X-ray are actively used for medical analysis. MRI is very frequently used in the quantitative assessment of brain-related disorders such as Alzheimer’s disease, epilepsy, schizophrenia, multiple sclerosis, and brain cancer [1]. The anatomical examination of the brain and determination of the anatomical regions is an important research area since conditions such as deformity in the anatomical parts can be a symptom of a disease. For this reason, detecting anatomical regions is frequently used in subjects such as anatomical development, computer-assisted detection, computer-assisted diagnosis, and patient follow-up [2].

Brain-specific anatomical studies are carried out to reveal many different structural and functional functions: extraction of the brain atlas [3], clipping areas that do not belong to the brain, such as the eye and skull, that are visible in MRI [4,5], revealing degenerations in anatomical parts [6], quantitative analysis of diseases [6], and estimation of brain age [7]. Bermudez et al. have shown that anatomical context and features increase the prediction success [7]. However, in some cases, anatomical regions such as the eye or skull visible on MRI may be evaluated as tumors by automated systems or cause bias. To avoid these negative effects, regions are detected in the preprocessing step and are not included in the analysis [6].

When studies on the detection of anatomical regions in the medical field are examined in detail, it is seen that early studies used traditional image processing techniques and generally aimed at enhancing image interpretation [8,9]. Then, studies began to propose machine learning and artificial intelligence-based solutions. These solutions mainly focused on object recognition, object detection, and segmentation. Heckemann et al. developed a prediction method for automatic brain MRI segmentation [10]. In the study of Tu et al., machine-learning-based discriminative/generative models were proposed for brain anatomical structure segmentation [11]. Bagci et al. proposed a hierarchical ball scale-based method for multiobject recognition in 3D anatomical structures [12]. They evaluated the technique they developed on both CT and MRI datasets. Cerrolaza et al. introduced a new hierarchical segmentation framework that used wavelet transform to segment anatomical structures containing multiple objects [13]. Bagci et al. developed a graph-based joint segmentation method that used anatomical and functional images together from a different perspective [14]. Brebisson and Montana presented a deep neural network-based novel proposal for anatomical brain segmentation [15]. Finally, Alansary et al. used deep reinforcement learning for anatomical landmark detection in brain MRI [16].

The segmentation task is frequently preferred in deep learning-based anatomical brain MRI studies. Segmentation is generally performed to extract normal regions (e.g., white matter, gray matter, and cerebrospinal fluid) [17] and abnormal regions (e.g., tumor and edema) [18]. One of the most important reasons for this situation is that the existing medical datasets are sufficient for only this much, both in the number and number of labels. In addition, it is imperative to detect anatomical structures that belong to the brain and do not belong to the brain, such as the eye and skull, because making this determination has direct effects on the high success rate of the automated systems. Segmentation studies are divided into three primary CNN-based groups based on the deep learning architecture used: patch-wise, semantic-wise, and cascaded architectures [4]. Patch-wise architectures are one of the primary approaches, and the training in these architectures is performed with NxN patches around the relevant pixel. In semantic-wise segmentation, each pixel is assigned to one of the predetermined class labels to make it meaningful. Lastly, in cascaded architectures, more than one CNN architecture is used together, and the output of the former is used as the latter’s input. In this context, Akil et al. performed fully automatic brain tumor segmentation for high-grade and low-grade glioblastoma with convolutional neural network CNN-based selective attention using overlapping patches and multiclass weighted cross entropy [19]. Liu et al. proposed an anatomical landmark-based deep feature learning framework for automatically extracting patch-based representation from brain MRI for Alzheimer’s disease [20]. Finally, Basher et al. suggested a two-stage ensemble CNN for real-time 3D anatomical structure localization from MRI [21].

Segmentation approaches are dominant in anatomical studies specific to brain MRI. One of the most important reasons for this is the characteristics of the publicly available datasets. However, it is seen that adding an object detection framework improves the performance and stability of brain tumor segmentation models [22]. Object detection describes the process of accurately representing relevant objects in an image in terms of their position, size, and shape definition. Object detection approaches are generally divided into two categories [23]: data-centered, such as edge detection and region growth; and goal-directed, such as knowledge-guided boundary detection. In this context, object detection offers great opportunities for disease diagnosis and treatment planning. However, object detection models are mostly designed for general images, and medical images differ significantly from general images due to features such as adverse viewing conditions, noise, and other artifacts. Therefore, object detection models may result in poor performance when applied directly to the medical field.

Ensemble techniques are one of the methods used to improve performance in brain MRI studies for several tasks as in many other fields. According to the current ensemble studies summarized in Table 1, the model ensemble can be achieved through two different techniques, namely feature-level ensemble and combining model outputs. In the feature-level ensemble technique, the features of various inputs are combined during model training; while in the model output combining technique, the outputs of multiple models are combined using various strategies without being limited to a single classifier/segmenter/regressor model, resulting in more accurate and reliable results. As seen in Table 1, these studies are used in segmentation, classification, and regression tasks in many areas such as age estimation, Alzheimer’s detection, tumor segmentation, and tumor classification.

In Aurna et al.’s study, a two-stage ensemble-based model was proposed for the tumor classification problem by combining the features extracted from deep learning architectures such as Custom CNN, EfficientNet-B0, and ResNet-50 and classifying them using classical machine learning methods such as support vector machine (SVM), random forest (RF), etc. [24]. Similarly, Aamir et al. also combined features extracted from EfficientNet and ResNet50 architectures for ensemble learning to perform tumor classification. Feature-level ensemble strategies have also been used for tumor segmentation tasks [25]. In Liu et al.’s study, an architecture called PIF-Net was proposed, based on the combination of features from different MRI modalities [26]. In Kua et al.’s study, brain age estimation with the scope of the regression task was performed using ridge regression and support vector regression (SVR) with ResNet [27].

In contrast to feature-level ensemble strategies, Dolz et al. improved brain tissue segmentation performance for infants by combining the results of customized 3D CNN variants using the majority voting method [28]. For the tumor segmentation task, Cabria et al. improved the performance by combining the results of the potential field segmentation, FOR, and PFC methods with rule-based [29], Feng et al. took the average of the results of 3D Unet variants [30], and Das et al. combined the results of the Basic Encoder–Decoder, U-Net, and SegNet models according to their success rates [31]. For the classification task, Tandel et al. combined the outputs of the AlexNet, VGG16, ResNet18, GoogleNet, and ResNet50 models using the majority voting method [32], while Islam et al. combined the outputs of the DenseNet121, VGG19, and Inception V3 models [33], and Ghafourian et al. combined the outputs of SVM, naive Bayes, and KNN [34] in a similar manner. In Kang et al.’s study, the results of multiple models were combined by taking their average values, resulting in improved tumor classification results [35]. For Parkinson’s detection, Kurmi et al. used a fuzzy logic-based ensemble strategy with the VGG16, ResNet50, Inception-V3, and Xception models [36]; while for Alzheimer’s detection, Chatter et al. combined the outputs of the SVM, logistic regression, naive Bayes, and K nearest neighbor methods using majority voting [37]. Finally, in Zahoor et al.’s study, both feature-level and model-level ensemble strategies were employed using multiple models for tumor classification [38].

Upon reviewing the literature, we observed that most of the ensemble studies were conducted for the tasks of tumor classification and segmentation. To the best of the author’s knowledge, there is currently no comprehensive ensemble study for the object detection task for brain MRI studies. The main contributions of this paper are as follows:A comprehensive ensemble-based object detection study for anatomical and pathological object detection in brain MRI.A total of nine state-of-the-art object detection models were employed to propose and evaluate four distinct ensemble strategies aimed at improving the accuracy and robustness of detecting anatomical and pathological regions in brain MRIs. The efficacy of these strategies was empirically assessed through rigorous experiments.A comparative evaluation of the current state-of-the-art object detection models for identifying anatomical and pathological regions in brain MRIs was conducted as a benchmarking study in the novel Gazi Brains 2020 dataset.Five different anatomical structures such as the brain tissue, eyes, optic nerves, lateral ventricles, and the third ventricle, as well as pathological objects including whole tumor parts seen in brain MRI, were detected simultaneously.

**Table 1 diagnostics-13-01494-t001:** Current studies in the brain MRI literature using ensemble strategies.

Ref.	Purpose	Methods	Ensemble Strategy	Learning Task	Year
[28]	Brain tissue segmentation for infant	Custom 3D CNN	Model outputs are combined using majority voting	Segmentation	2019
[29]	Tumor segmentation	Potential Field Segmentation, FOR, and PFC	Model outputs are combined by rule	Segmentation	2016
[32]	Tumor classification	AlexNet, VGG, ResNet, and GoogleNet	Model outputs are combined using majority voting	Classification	2022
[24]	Tumor classification	Custom CNN, EfficientNet-B0, and ResNet, Support Vector Machine, Random Forest, K-Nearest Neighbor, and AdaBoost.	Feature level ensemble	Classification	2022
[25]	Tumor classification	EfficientNet and ResNet	Feature level ensemble	Classification	2022
[37]	Alzheimer’s disease classification	SVM, Logistic Regression, Naive Bayes, and K-Nearest Neighbor	Model outputs are combined using majority voting	Classification	2022
[36]	Parkinson’s detection	VGG16, ResNet, Inception-V3, and Xception	Model outputs are combined using fuzzy logic	Classification	2022
[30]	Tumor segmentation and survival prediction	3D U-Net Variants	Model outputs are averaged	Segmentation	2020
[31]	Tumor segmentation	Basic Encoder–Decoder, U-Net, and SegNet model	Models outputs are combined based on accuracy	Segmentation	2022
[26]	Tumor segmentation	PIF-Net	Pixel and feature level ensemble	Segmentation	2022
[27]	Age estimation	Ridge Regression and Support Vector Regression (SVR) and Resnet	Feature level ensemble	Regression	2021
[33]	Tumor detection with federated learning	DenseNet, VGG, and Inception V3	Model outputs are combined using majority voting	Classification	2022
[35]	Tumor classification	ResNet, DenseNet, VGG, AlexNet, Inception V3, ResNext, ShuffleNet, MobileNet, MnasNet, FC layer, Gaussian NB, AdaBoost, K-NN, RF, SVM, and ELM	Model outputs are averaged	Classification	2021
[38]	Tumor classification	VGG, SqueezeNet, GoogleNet, ResNet, XceptionNet, InceptionV3, ShuffleNet, DenseNet, SVM, MLP, and AdaBoost	Features and classifiers ensemble	Classification	2022
[34]	Tumor classification	SVM, Naive Bayes, and K-NN	Model outputs are combined using majority voting	Classification	2023
This study	Anatomical and pathological object detection	RetinaNet, YOLOv3, FCOS, NAS-FPN, ATSS, VFNet, Faster R-CNN, Dynamic R-CNN, and Cascade R-CNN	Model bounding box outputs are recalculated using weighted sum	Object Detection	

## 2. Materials and Methods

### 2.1. Dataset

Datasets used in brain MRI studies are generally subject-specific, small, and non-public [39,40,41]. Researchers have difficulty in making the brain MRI data open for interdisciplinary study due to the necessity for domain experts to label this data, the time taken to label data, ethical issues for data sharing, etc. Therefore, although the number of medical images taken for disease detection and follow-up is relatively high, the conversion rate of these images into datasets is low.

The Gazi Brains 2020 dataset [42] was used in this study because it contains rich labeling information for both abnormal and normal patients, including various anatomical structures. The Gazi Brains 2020 dataset includes not only the anatomical parts of the brain but also anatomical parts such as the eye and optic nerve that can be seen in any brain MRI. In addition, considering the changes in anatomical parts such as the lateral and third ventricles, especially in the slices where the tumor is seen, it is a more challenging dataset in terms of finding these anatomical parts. The Gazi Brains 2020 dataset includes 50 normal and 50 histologically proven high-grade glioma (HGG) patients and has a total of 12 different types of label information provided by medical experts. All 100 patients have FLAIR, T1w, and T2w sequences, while 50 HGG groups and 12 normal patients have post-contrast T1 sequences. There are anatomical structures and pathological entities in the 12 different labels. In this study, the brain ROI (brain tissue and orbital CSF), eye, optic nerve, lateral ventricle, third ventricle, peritumoral edema, contrast-enhancing part, necrosis of tumor, hemorrhage, and the no contrast-enhancing part were used for the object detection models. The dataset statistics used in this study are given in Table 2. The labels for peritumoral edema, contrast-enhancing part, necrosis of tumor, hemorrhage, and no contrast-enhancing part were combined as the whole tumor objects.

In this study, the BraTS 2020 HGG dataset was also used for only pathological object detection. In total, 19,176 slice analyses with any tumor labels (NCR/NET—label 1, ED—label 2, ET—label 4) were used in the BraTS 2020 HGG dataset [39,43,44]. Slices without any labels were excluded from the analyses.

The dataset preparation process is visualized in Figure 1. The same dataset preparation process was performed as proposed by Terzi et al. [45]. Accordingly, each independent mask was defined as an object in a slice. Thus, there could be several different objects belonging to an anatomical structure in a slice. For example, in the top row of Figure 1b, there are several independent whole tumor masks, highlighted in white. Similarly, there are two independent lateral ventricles, the optic nerve, and the eye in the bottom row of Figure 1b. They were all taken as different objects, and these objects were used in the model training process as shown in Figure 1c.

### 2.2. Deep Learning Architectures for Anatomical and Pathological Object Detection

Object detection is basically evaluated under two models: one-stage and two-stage [46]. In two-stage techniques, first, the regions of objects are identified; then, the model is fed with region proposals for object classification and localization. Nonetheless, in one-stage techniques, a single model is applied that divides the image into regions and reveals the bounding box and label possibilities for each region.

In this study, all the models were trained using MMDetection [47], an object detection toolbox, to avoid various model implementation problems and to provide a standard training pipeline. A total of nine different state-of-the-art object detection models were used, as given below.
One-Stage Object Detection Models: RetinaNet [48], YOLOv3 [49], FCOS [50], NAS-FPN [51], ATSS [52], and VFNet [53];Two-Stage Object Detection Models: Faster R-CNN [54], Dynamic R-CNN [55], and Cascade R-CNN [56].

Two-stage object detection models consist of two stages: the region proposal network (RPN) and the detector. In the RPN stage, candidate regions are proposed, and in the detector stage, bounding box regression and classification are performed. Two-stage object detection models are also known as the R-CNN family, and there are many examples of this type of model.

Faster R-CNN is an improved version of the Fast R-CNN model that has a faster run time. To realize this, it uses a CNN-based feature extractor for proposing rectangular objects instead of a selective search in the proposal stage. The proposed object features are shared with the detector and used for bounding box regression and classification. A bounding box is defined as four different coordinates $b=\left(b_{x},b_{y},b_{w},b_{h}\right)$, and the regression is performed using the smoothed L1 loss function (Equation (Equation 2)) between the ground truth bounding box and the candidate bounding box. The loss function learns the coordinates from the data by trying to minimize the distance. For the classification process, the classifier performs learning by minimizing the classification cross-entropy loss (Equation (Equation 1) for binary cross-entropy) on the training set. During the learning process, positive and negative detections are identified based on the IoU metric according to Equation (Equation 3)
(1)$$\begin{array}{c} p_{l}=\left\{\begin{array}{ll} p & \;\mathrm{if}\;y=1\\ 1-p & \;\mathrm{otherwise},\; \end{array}\right.\\ \mathrm{CE}\left(p,y\right)=\mathrm{CE}\left(p_{t}\right)=-\log\left(p_{t}\right), \end{array}$$
where *p* represents the class probability, and *y* represents the ground truth label.
(2)$${smooth}_{L_{1}}\left(x\right)=\left\{\begin{array}{ll} 0.5x^{2} & \;\mathrm{if}\;|x|<1\\ |x|-0.5 & \;\mathrm{otherwise}\; \end{array}\right.,$$
where *x* represents the regression label,
(3)$$label=\left\{\begin{array}{ll} 1, & \;\mathrm{if}\;\max IoU\left(b,G\right)\geq T_{+}\\ 0, & \;\mathrm{if}\;\max IoU\left(b,G\right)<T_{-}\\ -1, & \;\mathrm{otherwise}.\; \end{array}\right.$$

Here, *b* represents the bounding box, *G* represents the ground truth, and $T_{+}$ and $T_{-}$ represent the positive and negative thresholds for the IoU, respectively.

The Cascade R-CNN performs object detection by taking into account different IoU overlap levels compared to Faster R-CNN. The Cascade R-CNN architecture proposes a series of customized regressors for different IoU overlap threshold values, as given in Equation (Equation 4). *N* represents the total number of cascade stages, and *d* denotes the sample distribution in the equation.
(4)$$f\left(x,d\right)=f_{N}\circ f_{N-1}\circ \cdots \circ f_{1}\left(x,d\right)$$

The Dynamic R-CNN involves a dynamic training procedure compared to Faster and Cascade R-CNN. In this method, instead of using predefined IoU threshold values in the second stage, where the regressor and classifier are used, a dynamically changing procedure based on the distribution of region proposals is proposed given in Equation (Equation 3). For this purpose, the dynamic label assignment (DLA) process given in Equation (Equation 5) is first used according to the T threshold value updated based on the statistical distribution of proposals for object detection. For the localization task (bounding box regression), the Dynamic Smooth L1 Loss given in Equation (Equation 6) is used. Similarly to the DLA, the DSL also changes the regression labels based on the statistical distribution of proposals.
(5)$$label=\left\{\begin{array}{ll} 1, & \;\mathrm{if}\;\max IoU\left(b,G\right)\geq T_{\mathrm{now}\;}\\ 0, & \;\mathrm{if}\;\max IoU\left(b,G\right)<T_{\mathrm{now}\;} \end{array}\right.$$
where $T_{\mathrm{now}\;}$ represents the current IoU threshold and updates according to the data distribution,
(6)$$DSL\left(x,\beta_{\mathrm{now}\;}\right)=\left\{\begin{array}{ll} {0.5|x|}^{2}/\beta_{\mathrm{now}\;}, & \;\mathrm{if}\;|x|<\beta_{\mathrm{now}\;}\\ \left|x\right|-0.5\beta_{\mathrm{now}\;}, & \;\mathrm{otherwise}\; \end{array}\right.,$$
where $\beta_{\mathrm{now}\;}$ represents the hyperparameters to control smooth loss.

The main difference between one-stage object detection models and two-stage object detection models is that one-stage models do not involve a region proposal stage. Therefore, they generally provide faster detector performance compared to the two-stage methods. One-stage object detection models show diversity in many aspects. These models can be divided into various subcategories based on the differences in their architectures (such as anchor-based, anchor-free, feature pyramid, context, etc.). The following is a summary of the one-stage object detectors used in this study.

The most important component of the RetinaNet architecture is a loss function called Focal Loss, given in Equation (Equation 7). This proposed loss function solves the problem of class imbalance during training. It allows the rarer class to be learned better than the classic cross-entropy loss (Equation (Equation 1)), leading to a balanced model training. The focal loss equation is defined as the focal loss. RetinaNet is an anchor-based architecture that uses the Resnet-based FPN as its backbone. While the backbone is used to extract the convolutional features from the input, two different subnets that take the backbone’s outputs as inputs are used for object classification and bounding box regression.
(7)$$FL\left(p_{t}\right)=-\alpha_{t}{\left(1-p_{t}\right)}^{\gamma }\log\left(p_{t}\right)$$

YoloV3 is a variant of Yolo, which is an improved version of YoloV2 in various aspects. The most important factor that makes Yolo models faster than other object detection models is that they do not contain a complex pipeline. In this architecture, the entire image is used as the input to the network, and bounding box regression and object detection are directly performed on this image. The main differences between YoloV3 and YoloV2 are that YoloV3 uses a deeper architecture called Darknet-53 for feature extraction and predicts bounding boxes in three different scales.

FCOS is an anchor-free object detection model that performs object classification and bounding box regression without requiring overlap (IoU) calculations with anchors or detailed hyperparameter tuning. Anchor-based methods consider the center of anchor boxes as the location in the input image and use it to regress the target bounding box. However, FCOS considers any point inside the ground truth box as a location and uses the distance $t^{*}=\left(l^{*},t^{*},r^{*},b^{*}\right)$ (given in Equation (Equation 8)) for regression. If a point is inside multiple bounding boxes, it is considered an ambiguous example, and multilevel prediction is used to reduce this situation. FCOS also uses the centeredness strategy to improve the bounding box detection quality (given in Equation (Equation 9)). To do this, FCOS adds a parallel stage to the classification task to predict whether a location is a center or not. The centeredness value is trained using binary cross-entropy loss and added to the FCOS’s loss function.
(8)$$\begin{array}{cc} l^{*} & =x-x_{0}^{\left(i\right)},\;t^{*}=y-y_{0}^{\left(i\right)},\\ r^{*} & =x_{1}^{\left(i\right)}-x,\;b^{*}=y_{1}^{\left(i\right)}-y, \end{array}$$
where $l^{*},t^{*},r^{*},$ and $b^{*}$ are the distances calculated from the bounding box.
(9)$$\;\mathrm{centeredness}\;{}^{*}=\sqrt{\frac{\min\left(l^{*},r^{*}\right)}{\max\left(l^{*},r^{*}\right)}\times \frac{\min\left(t^{*},b^{*}\right)}{\max\left(t^{*},b^{*}\right)}}.$$

The NAS-FPN model has made improvements to the FPN model architecture used in object detection models for extracting visual representations. Unlike other models that use manual FPN architecture design, this model narrows down a wide search space for FPN architecture through the neural architecture search algorithm to extract the effective FPN architectures.

The ATSS model proposes a method that performs adaptive sampling of training set examples by utilizing the statistical characteristics of positive and negative examples, based on the fact that they have a significant impact on the model’s performance. This approach has led to significant improvements in both anchor-based and anchor-free detectors, filling the performance gap between these two different architectures.

VarifocalNet proposed a different approach compared to the other models in evaluating candidate objects during training by considering not only a classification score or a combination of classification and localization scores but also an IoU-based classification score (IACS) that takes detection performance into account. This approach takes both the localization accuracy and the object confidence score into consideration, resulting in successful results, especially in dense object detection scenarios. To estimate the IACS score, the varifocal loss (given in Equation (Equation 10)) and star-shaped bounding box representation were proposed.
(10)$$VFL\left(p,q\right)=\left\{\begin{array}{cc} -q(q\log(p)+(1-q)\log(1-p)) & q>0\\ -\alpha p^{\gamma }\log\left(1-p\right) & q=0, \end{array}\right.$$
where *p* represents the predicted IACS score, and *q* represents the target score.

### 2.3. Model Ensemble

The ensemble methods include combining the results of many models belonging to an object detection architecture, as well as combining the results of models with different architectures. In this study, four different ensemble strategies were presented as follows:Strategy 1: an ensemble of all cross-validated folds for a model;Strategy 2: an ensemble of the best folds for a model;Strategy 3: an ensemble of different models fold-by-fold;Strategy 4: an ensemble of the best different models.

In Figure 2, an ensemble method is presented in which the deep learning architecture is kept constant, and the predictions of many model variants belonging to the relevant architecture are combined to realize this ensemble strategy. Nine different object detection models with tenfold cross-validation were used to measure the efficiency of the ensemble strategy for each model. The numbers indicated in Figure 2 (e.g., A.1, A.10, etc.) represent the fold number of the relevant models. In Figure 2(a.1), the predictions of all model variants belonging to the tenfold cross-validated Model A are combined without any fold selection criteria. In Figure 2(a.2), instead of the ensemble of all the tenfold predictions, the predictions of the best top-k cross-validated models for Model A are combined with the best model selection criteria. In this study, the top-k models were selected as the folds that produced an mAP above the average mAP value, which came from the tenfold cross-validation. Ensemble strategies for a relevant model produce a single ensemble result for all and the top-k folds.

An ensemble strategy of different models is presented in Figure 2b. In the figure, unlike the ensemble strategy in Figure 2a, the fold number is kept constant, and the predictions of the different models are combined. In Figure 2(b.3), the predictions of the models that come from tenfold cross-validation for each different model are combined fold-by-fold without any selection criteria. This ensemble strategy generated a total of ten ensemble results, with one ensemble result per fold. In Figure 2(b.4), for each fold, the best model among all models is selected, and the best models corresponding to each fold are combined to produce a single ensemble result in this strategy.

Predictions of the models and the bounding boxes for each object were combined using the weighted boxes fusion (WBF) [57] method for all ensemble strategies. For the WBF, firstly, the confidence score was calculated for each model as in Equation (Equation 11). The coordinates of the new bounding boxes were recalculated with the help of Equations (Equation 12) and (Equation 13). In the equations, C represents the confidence score, T represents all boxes, and X and Y represent the new bounding box coordinates. In this study, the confidence score for each model prediction was used as the average score of the intersecting bounding boxes. The new coordinates were calculated using the weighted sum of the confidence scores of the boxes. Thus, the effectiveness of the boxes with higher confidence scores was increased.
(11)$$C=\frac{\sum_{i=1}^{T}C_{i}}{T}$$
(12)$$X1,2=\frac{\sum_{i=1}^{T}C_{i}*X1,2_{i}}{\sum_{i=1}^{T}C_{i}}$$
(13)$$Y1,2=\frac{\sum_{i=1}^{T}C_{i}*Y1,2_{i}}{\sum_{i=1}^{T}C_{i}}$$

## 3. Results

### 3.1. Evaluation Metrics

Object detection aims to detect objects in an image, their classes, and their locations. For this reason, the object detection task has its own evaluation metrics. Two different popular object detection metrics, the Intersection over Union (*IoU*) and the mean average precision (*mAP*), were used to evaluate the results of this study.

Considering the ground truth (*A*) and the predicted bounding box (*B*) as two different clusters, in this situation, the ratio of the intersection of these two clusters to the union represents the *IoU* [58] value (Equation (Equation 14)).
(14)$$IoU=\frac{\left|A\cap B\right|}{\left|A\cup B\right|}=\frac{\left|A\cap B\right|}{\left|A|+|B|-|A\cap B\right|}$$

Precision (*P*) and Recall (*R*), which are classical machine learning metrics, are used as the basis for calculating the *mAP* value. As can be seen from Equations (Equation 15) and (Equation 16), *P* indicates the model’s accuracy in classifying a sample as positive, while *R* indicates the power of the model to detect positive samples. In the equations, *TP* (True Positive) refers to the model correctly detecting the object. *FN* (False Negative) means the object cannot be detected by the model, although there is an object in the image. *FP* (False Positive) means an object is detected, even though there is no object present. In the object detection task, these metrics are calculated according to the case that the *IoU* value between the predicted boxes and the ground truth boxes is above a certain threshold value.
(15)$$P=\frac{TP}{TP+FP}$$
(16)$$R-Recall-Sensitivity\left(TPR\right)=\frac{TP}{TP+FN}$$

Therefore, these two metrics are also important, and for this reason, the P–R curve was used, which shows the balance between the P and R for different thresholds. The average precision (*AP*) summarizes the P–R curve and was calculated according to Equation (Equation 17) ($R_{n}$ = 0, $P_{n}$ = 1, n = the number of thresholds). After calculating the *AP* value for each class, the *mAP* value was found by taking the average of them [59]. In this study, the FROC curves were plotted using the *FPPI* (Equation (Equation 18)) and *TPR* (Equation (Equation 16)) to measure how changes in the confidence score affected both the False Positive rate per image and the *TPR* performance.
(17)$$AP=\sum_{k=0}^{k=n-1}\left(R_{k}-R_{k+1}\right)P_{k}$$
(18)$$FPPI=\frac{FP}{ImageNumbers}$$

### 3.2. Experimental Setup

In this study, 2029 axial brain MRI slices belonging to 100 patients in the Gazi Brains 2020 dataset were separated on a patient basis. First, 202 slices belonging to 10 patients including five normal and five tumorous patients (Patient IDs: Sub-10, 14, 15, 19, 32, 53, 59, 61, 84, and 94) were selected randomly for testing purposes only. Then, 1827 slices belonging to the remaining 90 patients were divided into training and validation sets according to the tenfold cross-validation method. Each fold included 1644 (±5) training slices and 182 (±5) validating slices. In each fold, the model with the best mAP value in the validation set was selected and tested. This process was repeated for all folds, and benchmarks were made to report. For the BraTS 2020 HGG dataset, the dataset was divided on a patient basis into 60% training, 20% validation, and 20% testing, randomly. All models were selected based on the validation performance and then tested for reporting.

The FLAIR, T1w, and T2w sequences, which were spatially aligned and common sequences for all patients as three channels, were prepared for the model training. The backbone, hyperparameters, and augmentation techniques used in the training process of the models are given in Table 3. All layers of the model were trained with the initial weights coming from the pretrained models that were obtained from the MMDetection GitHub repository. All augmentation techniques were applied with 0.5 probability, and finally, a multiscale flip augmentation technique was applied for the testing pipeline. The default values came from the MMDetection repository and were used for all the parameters except the hyperparameters specified in Table 3.

During the model testing process, non-maximum suppression (NMS) was applied to the model predictions by using 0.6 IoU and 0.1 score threshold values. After applying the NMS to the predictions, the WBF was applied for the ensemble strategies. The WBF weights were taken equally as 1. Moreover, 0.6 IoU and 0.16 skip boxes threshold were used as the WBF hyperparameters.

### 3.3. Experimental Results

In this section, two different benchmark results are given for the anatomical and pathological object detection problem on the Gazi Brains 2020 dataset. The first one aims to present the performance of each model individually, and the second one aims to present the ensemble performance of the models within the scope of the determined strategies.

The average values of the tenfold cross-validation results on the test set for each model are given in Table 4. As can be seen from the table, the best class-based AP(@0.5 IoU) values varied according to the models. In terms of the detection of each anatomical object, the benchmarking results were summarized as follows:All the models had similar success results for the brain ROI object;The Dynamic R-CNN had a 0.95 AP(@0.5 IoU) value for the eye object;The RetinaNet and VFNet models had a 0.59 AP(@0.5 IoU) value for the optic nerve object;The NAS-FPN and YOLOv3 models had a 0.62 AP(@0.5 IoU) value for the third ventricle object;The NAS-FPN and RetinaNet had a 0.82 AP(@0.5 IoU) value for the tumor object;The most successful model according to the mean average precision was the NAS-FPN with a 0.76 (±0.02) (@0.5 IoU) mAP value.

The ensemble results of different variants for each model are given in Table 5. To realize this strategy, the models from the tenfold cross-validation of the relevant model were used, and the efficiency of the ensemble strategies was reported compared to the best model. In the table, the best fold according to the mAP value (the Best Fold) is compared to the ensemble outputs of the folds above the average mAP (Mean+ Folds, Figure 2(a.1)) and all folds without any model selection criteria (All Folds, Figure 2(a.2)). According to the results reported in Table 5, the ensemble of the models increased the anatomical object detection success rate for each model. Accordingly, the ensemble strategies produced better mAP values by 2% for ATSS, 3% for Cascade R-CNN, 4% for Dynamic R-CNN, 2% for Faster R-CNN, 1% for FCOS, 2% for NAS-FPN, 3% for RetinaNet, 2% for VFNet, and 10% for YOLOv3. The ensemble strategies performed with the models’ own variants provided improvements in the better detection of the different anatomical objects for each model. The most interesting and satisfactory results obtained were 5%, 9%, 11%, 6%, 7%, and 18% improvements observed in finding the optic nerve object for the ATSS, Cascade R-CNN, Dynamic R-CNN, Faster R-CNN, VFNet, and YOLOv3 models, respectively. Similarly, 5% and 8% improvements were observed in the NAS-FPN and RetinaNet models, while a 3% improvement in finding the tumor object was observed in the FCOS model.

The ensemble results of different models are given in Table 6. In the table, the results obtained from the fold-by-fold ensemble strategy are given for each fold (All Folds- 1, 2,…,10). In addition, the average results of the fold-by-fold ensemble strategy (All Folds—Mean, Figure 2(b.3)), the results of the best fold according to the mAP value in this strategy (All Folds–the Best, Figure 2(b.3)), and the ensemble results obtained by selecting the best model for each fold (the Best Folds, Figure 2(b.4)) were compared.

The ensemble of all the models in any fold gave better results than the best model in any fold. For example, the best model of the NAS-FPN produced an mAP value of 0.805, as indicated in Table 5. However, when the values in Table 6 were examined together, the different model ensemble strategy was more successful in all the folds except Fold 9. Similarly, an ensemble of the different models based on the tenfold cross-validation, with a 0.818 (±0.01) mAP, yielded about 6% better mAP results than the best individual model, which is indicated in Table 4, the NAS-FPN with a 0.76 (±0.02) mAP value.

Finally, an ensemble of the different best model variants with a 0.838 mAP value as indicated in Table 6 was 1.8% and 1% better than the other ensemble strategies, such as the ensemble of the NAS-FPN model variants with a 0.82 mAP value in Table 5 and an ensemble of the best different models fold-by-fold with a 0.828 mAP value in Table 6. It was also 3.3% better than the individual NAS-FPN best models with a 0.805 mAP value in Table 5.

The comparative FROC curves with the FAUC values representing the area under these curves are provided in Figure 3 for each anatomical and pathological region using the best individual models and the ensemble strategy (Strategy 4—Best Folds) specified in Table 6. Thus, the effect of the changes in the confidence scores ranging from 0 to 1 with intervals of 0.02 on the FPPI and TPR were measured on the Gazi Brains 2020 dataset. According to the results provided in the figure, the ’Best Folds Ensemble’ strategy outperformed with an approximately 1%, 2%, 1%, 2%, and 7% better FAUC performance for the RoI, optic nerve, eye, lateral ventricle, and third ventricle, respectively, compared to the best individual model.

In this study, experiments were also conducted on the benchmark of the models and the ensemble strategy of the best models for detecting the pathological regions on the BraTS 2020 HGG dataset. The anatomical regions were not considered because the BraTS 2020 HGG dataset lacks labels for the anatomical regions. When considering the AP metric with a 0.2 confidence score and a 0.5 IoU threshold for a tumor, the ATSS, Cascade R-CNN, Dynamic R-CNN, Faster R-CNN, FCOS, Nas-FPN, RetinaNET, VfNet, YOLOV3, and the ensemble strategy of different models produced AP values of 0.77, 0.782, 0.784, 0.783, 0.774, 0.788, 0.8, 0.791, 0.742, and 0.843, respectively. The ensemble strategy outperformed the best individual model by approximately 4%. Furthermore, Figure 4 presents the FROC curves of the models based on their TPR and FPPI values, considering different confidence scores. The ensemble strategy of different models outperformed with an approximately 2% better FAUC performance than the best individual model (Faster R-CNN).

Figure 5 visualizes the efficiency of the model ensemble. For visualization purposes, we selected the best ATSS, NAS-FPN, Cascade R-CNN, Dynamic R-CNN, Faster R-CNN, FCOS, and YOLOv3 models from left to right, respectively. Moreover, the classification confidence threshold was selected as 0.3 for each class. As shown in the figure, the ensemble results using different models found anatomical parts that could not be found by a single model and reduced unnecessary predictions and false positives.

## 4. Discussion

Automatic anatomical object detection models based on deep learning, as a computer-aided diagnostic tool, support the decision-making processes of clinicians by presenting the location, shape, and size of the object in the medical image. However, sometimes the performance of these models cannot meet clinical expectations due to some issues arising from medical images. Anatomical objects do not always differentiate well in contrast and may appear similar to neighboring structures. Even if the characteristics of an anatomical object are known, they often vary from case to case. At the same time, most different types of objects have similar characteristics in size and shape. In addition, noise and artifacts greatly alter the depicted objects. On the other hand, although valuable information is gained about anatomy, extensive expert knowledge is needed to fill the semantic gap between the depicted objects and the image data. Annotating is time-consuming and expensive, as this process involves experts interpreting the medical images and combining them with other test results if necessary. Due to all these problems, medical object detection models may not be as robust as expected.

As in many of the studies listed in Table 2, this study was also conducted with a single dataset. This can lead to overfitting risks for deep learning models developed on limited datasets, limiting their generalization abilities. Although it may be possible to combine data from different open sources, finding datasets with similar labels or similar tasks is almost impossible. In this study, the performance of models was measured using the tenfold cross-validation method on the Gazi Brains 2020 open dataset and the BraTS 2020 HGG dataset. As expected, ensemble strategies are more effective in relatively small datasets such as Gazi Brains 2020, but they also prove to be effective in larger datasets such as BraTS 2020 HGG. This is particularly important in situations where data are limited, highlighting the necessity of ensemble strategies.

Therefore, this study aimed to combine decisions from multiple models using different ensemble strategies and to improve anatomical object detection model performance. In this study, nine different models were compared for the anatomical object detection problem, and four different ensemble strategies were proposed. The proposed ensemble strategies ensured that each model individually boosted the anatomical object detection capacity and achieved the highest success rates by combining different models.

## 5. Conclusions

According to the obtained results, the ensemble strategies performed on the model variants were found to change the individual performance of the models, improving the mAP values between 1% and 10% compared to the relevant best model. Similarly, when looking at the changes in the detection performance of the anatomical parts, an improvement of up to 18% AP was observed. In particular, ensemble strategies were found to be much more efficient in determining anatomical parts with a small amount of data, such as the optic nerve and third ventricle. As seen in Table 5, the highest performance increase in eight of the nine models was for the optic nerve and third ventricle anatomical objects. It was also observed that the ensemble of different models outperformed the individual best model in almost all cases, and the ensemble of the different best models was 3.3% better than the best individual model, which was the NAS-FPN. Another important output of the ensemble strategies was that, as seen in Figure 3 and Figure 4, they produced higher TPR values, especially at low FPPI values, compared to the best individual methods.

It is planned to implement ensemble strategies for different tasks such as classification and segmentation in future studies. It is also planned to use anatomical object detection methods in the cascaded architectures for extracting better anatomic patches to realize better segmentation tasks.

## Figures and Tables

**Figure 1 diagnostics-13-01494-f001:**
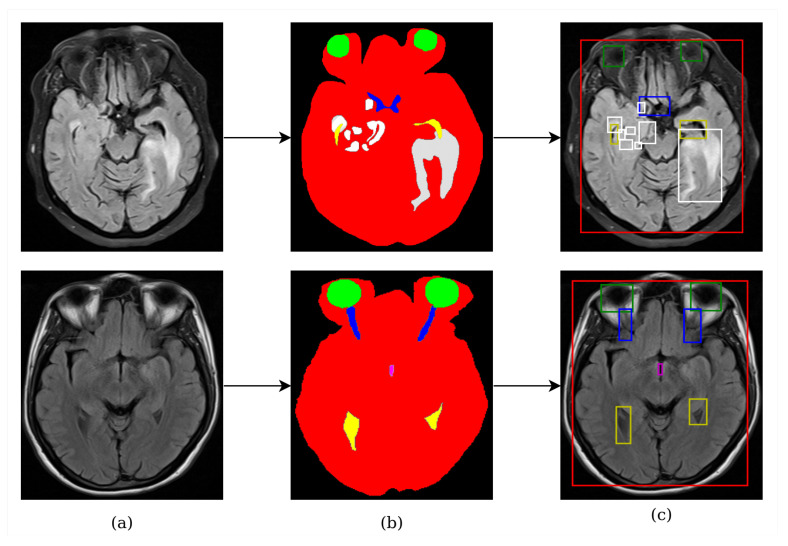
An example of the dataset preparation: (**a**) an original FLAIR slice; (**b**) segmented by experts; and (**c**) the extracted anatomical objects (red: ROI, green: eye, blue: optic nerve, yellow: lateral ventricle, magenta: third ventricle, white: whole tumor).

**Figure 2 diagnostics-13-01494-f002:**
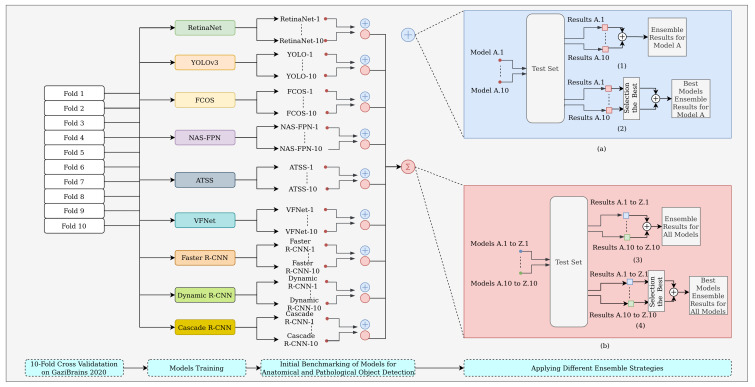
Workflow of the proposed methodology and ensemble strategies for anatomical and pathological object detection: (**a**) An ensemble of different variants of a model; (1) for all folds, Strategy 1, and (2) for the best folds, Strategy 2. (**b**) An ensemble of different models fold-by-fold; (3) for all models, Strategy 3, and (4) for the best different models, Strategy 4.

**Figure 3 diagnostics-13-01494-f003:**
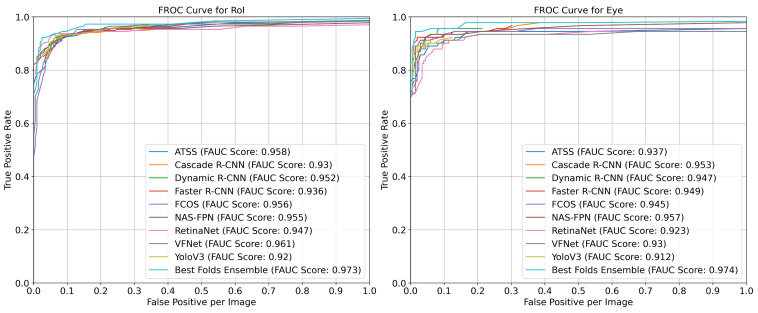
FROC curves based on the TPR and FPPI for each anatomical and pathological region for the best ensemble strategy and the best individual models on the Gazi Brains 2020 dataset.

**Figure 4 diagnostics-13-01494-f004:**
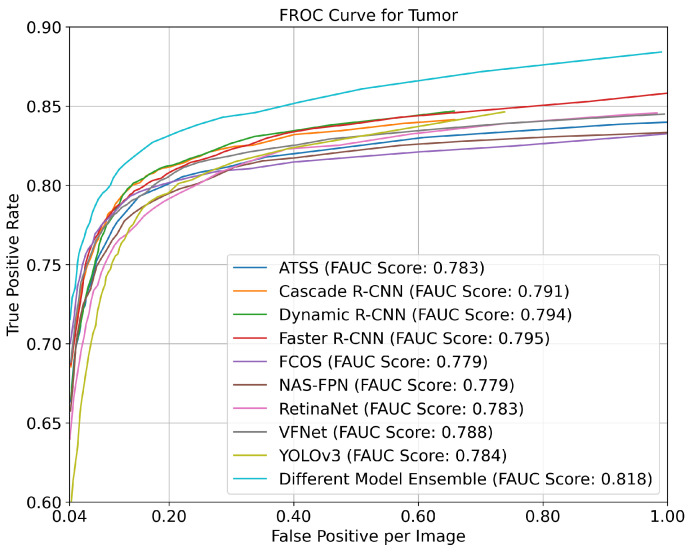
FROC curve for the pathological region for the best different model strategy on the BraTS 2020 Dataset.

**Figure 5 diagnostics-13-01494-f005:**
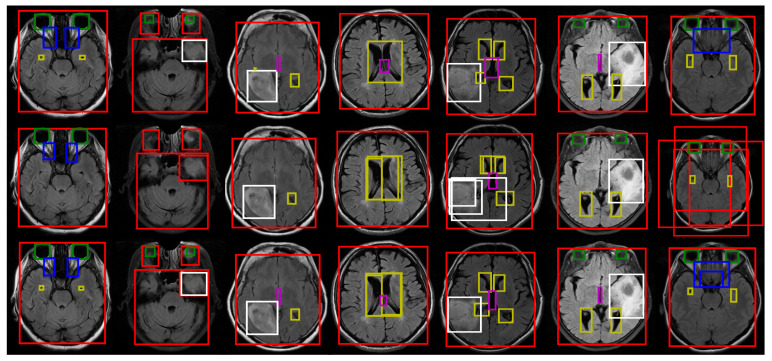
Example results for the best different models ensemble strategy on the FLAIR sequence. The first row shows the ground truth, the second row shows the best individual model, and the third row shows the ensemble results for the different best models. (red: ROI, green: eye, blue: optic nerve, yellow: lateral ventricle, magenta: third ventricle, white: whole tumor).

**Table 2 diagnostics-13-01494-t002:** Gazi Brains 2020 Dataset Statistics.

**Total Patients**	100 patients (50 normal, 50 HGG)
**Total Number of Slices**	2209
**Number of Anatomical and Pathological Objects**	Brain ROI: 2209 Eye: 476 Optic Nerve: 233 Lateral Ventricle: 789 Third Ventricle 388 Peritumoral Edema: 427 Contrast Enhancing Part: 309 Tumor Necrosis: 221 Hemorrhage: 23 No Contrast Enhancing Part: 77
**Number of Structures Seen in Slices**	Brain ROI: 2628 Eye: 928 Optic Nerve: 357 Lateral Ventricle: 1988 Third Ventricle: 403 Whole Tumor: 556

**Table 3 diagnostics-13-01494-t003:** Hyperparameter settings for the models.

Models	Backbone	Common Hyperparameters
ATSS Cascade R-CNN Dynamic R-CNN Faster R-CNN FCOS NAS-FPN RetinaNet VFNet YOLOv3	ResNet_101 ResNext_101 ResNet_50 ResNext_101 ResNext_101 ResNet_50 with RetinaNet ResNext_101 ResNext_101 Darknet_53	sample_per_gpu:4 workers_per_gpu:2 image_size:512*512 optimizer:SGD lr_rate:5e-3 lr_config:linear step epoch_num:50 gpu_num:4 augmentation: random brightness (0.15) contrast (0.15) vertical flip

**Table 4 diagnostics-13-01494-t004:** Class-based average precision results with tenfold cross-validation on the test set.

Model	Brain ROI	Eye	Optic Nerve	Lateral Ventricle	Third Ventricle	Whole Tumor	mAP
ATSS	0.96 (±0.01)	0.92 (±0.02)	0.57 (±0.04)	0.64 (±0.02)	0.56 (±0.05)	0.79 (±0.02)	0.74 (±0.01)
Cascade R-CNN	0.96 (±0.01)	0.93 (±0.02)	0.55 (±0.04)	0.63 (±0.02)	0.51 (±0.08)	0.79 (±0.03)	0.73 (±0.02)
Dynamic R-CNN	0.96 (±0.01)	0.95 (±0.01)	0.40 (±0.06)	0.66 (±0.01)	0.45 (±0.06)	0.81 (±0.02)	0.71 (±0.01)
Faster R-CNN	0.95 (±0.01)	0.94 (±0.01)	0.48 (±0.05)	0.64 (±0.01)	0.45 (±0.06)	0.79 (±0.03)	0.71 (±0.02)
FCOS	0.96 (±0.01)	0.94 (±0.01)	0.54 (±0.07)	0.65 (±0.01)	0.55 (±0.06)	0.77 (±0.04)	0.73 (±0.02)
NAS-FPN	0.96 (±0.01)	0.93 (±0.02)	0.58 (±0.08)	0.64 (±0.02)	0.62 (±0.05)	0.82 (±0.04)	0.76 (±0.02)
RetinaNet	0.96 (±0.01)	0.91 (±0.01)	0.59 (±0.04)	0.61 (±0.03)	0.49 (±0.05)	0.82 (±0.02)	0.73 (±0.01)
VFNet	0.96 (±0.01)	0.92 (±0.01)	0.59 (±0.06)	0.66 (±0.02)	0.58 (±0.05)	0.78 (±0.03)	0.75 (±0.02)
YOLOv3	0.74 (±0.03)	0.90 (±0.01)	0.44 (±0.05)	0.57 (±0.02)	0.62 (±0.06)	0.74 (±0.03)	0.67 (±0.01)

**Table 5 diagnostics-13-01494-t005:** Ensemble Strategy 1 and 2 results for each model.

Model	Ensemble Strategy	Brain ROI	Eye	Optic Nerve	Lateral Ventricle	Third Ventricle	Whole Tumor	mAP	Dif.
ATSS	Best Fold	0.97	0.91	0.61	0.66	0.64	0.81	0.765	-
	Mean+ Folds	0.97	0.94	0.64	0.67	0.68	0.81	0.784	0.02
	All Folds	0.97	0.94	0.66	0.67	0.65	0.82	0.785	0.02
Cascade R-CNN	Best Fold	0.94	0.95	0.58	0.60	0.64	0.81	0.754	-
	Mean+ Folds	0.98	0.95	0.66	0.68	0.61	0.82	0.783	0.03
	All Folds	0.97	0.95	0.67	0.68	0.60	0.82	0.783	0.03
Dynamic R-CNN	Best Fold	0.96	0.94	0.45	0.67	0.50	0.82	0.723	-
	Mean+ Folds	0.97	0.94	0.56	0.67	0.57	0.84	0.757	0.03
	All Folds	0.97	0.96	0.52	0.69	0.59	0.83	0.760	0.04
Faster R-CNN	Best Fold	0.96	0.93	0.57	0.63	0.54	0.79	0.738	-
	Mean+ Folds	0.97	0.96	0.63	0.66	0.53	0.83	0.762	0.02
	All Folds	0.98	0.96	0.61	0.68	0.53	0.81	0.760	0.02
FCOS	Best Fold	0.96	0.93	0.65	0.65	0.61	0.78	0.762	-
	Mean+ Folds	0.96	0.95	0.58	0.66	0.55	0.80	0.749	-0.01
	All Folds	0.97	0.94	0.65	0.67	0.58	0.81	0.768	0.01
NAS-FPN	Best Fold	0.97	0.95	0.71	0.66	0.69	0.86	0.805	-
	Mean+ Folds	0.98	0.95	0.69	0.70	0.69	0.86	0.812	0.01
	All Folds	0.98	0.95	0.68	0.71	0.74	0.87	0.820	0.02
Retinanet	Best Fold	0.96	0.91	0.61	0.61	0.54	0.83	0.745	-
	Mean+ Folds	0.97	0.91	0.61	0.66	0.59	0.82	0.761	0.02
	All Folds	0.98	0.92	0.65	0.65	0.62	0.81	0.770	0.03
VFNet	Best Fold	0.97	0.92	0.62	0.66	0.67	0.79	0.772	-
	Mean+ Folds	0.98	0.93	0.69	0.70	0.66	0.82	0.793	0.02
	All Folds	0.98	0.95	0.67	0.70	0.65	0.81	0.792	0.02
YOLOv3	Best Fold	0.75	0.90	0.43	0.57	0.70	0.74	0.683	-
	Mean+ Folds	0.81	0.93	0.61	0.66	0.80	0.82	0.772	0.09
	All Folds	0.84	0.94	0.60	0.67	0.85	0.83	0.787	0.10

**Table 6 diagnostics-13-01494-t006:** Ensemble Strategy 3 and 4 results for different models fold-by-fold.

Ensemble Strategy	Brain ROI	Eye	Optic Nerve	Lateral Ventricle	Third Ventricle	Whole Tumor	mAP
All Folds - 1	0.98	0.95	0.67	0.73	0.72	0.87	0.820
All Folds - 2	0.98	0.97	0.70	0.71	0.74	0.85	0.825
All Folds - 3	0.98	0.96	0.68	0.71	0.67	0.86	0.810
All Folds - 4	0.98	0.97	0.73	0.71	0.73	0.85	0.828
All Folds - 5	0.98	0.96	0.67	0.70	0.77	0.85	0.821
All Folds - 6	0.98	0.96	0.68	0.71	0.72	0.85	0.815
All Folds - 7	0.98	0.98	0.69	0.69	0.75	0.85	0.824
All Folds - 8	0.98	0.97	0.67	0.71	0.76	0.86	0.822
All Folds - 9	0.98	0.96	0.68	0.71	0.64	0.86	0.804
All Folds - 10	0.98	0.95	0.70	0.69	0.67	0.85	0.808
Mean	0.98	0.96	0.69	0.71	0.72	0.86	0.818
the best (Fold-4)	0.98	0.97	0.73	0.71	0.73	0.85	0.828
Best Folds *	0.98	0.97	0.75	0.73	0.73	0.87	0.838

* The NAS-FPN for Folds 1, 3, 4, and 7; FCOS for Fold 2 and 6; VFNet for Fold 5 and 8; ATSS for Fold 9; and Cascade R-CNN for Fold 10 were selected as the best models for the ensemble strategy.

## Data Availability

The Gazi Brains 2020 dataset is available in BIDS format Synapse Repo at https://www.synapse.org/#!Synapse:syn22159468/wiki/603890, accessed on 23 January 2023. The BraTS 2020 HGG dataset can be found in [39,43,44].

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
