# Peer review of "An Ensemble of Deep Learning Object Detection Models for Anatomical and Pathological Regions in Brain MRI"

_diagnostics, 2023, doi:10.3390/diagnostics13081494_

Round 1
Reviewer 1 Report
I have the following problems with this paper:
- The problem formulation is skipped. It simply narrates the previous work without mentioning the similarities and differences between this method and previous methods, and the innovation is not prominent.
- It is difficult for readers to understand the main contributions of this paper. The Introduction can be revised to emphasize the main contribution of the work.
- The article has poor math formulating and a weak technical description.
- The architecture of the framework needs to be described in more detail. The training process and how the parameters are set must be deepened.
- I would suggest the authors present more graphical information. It will be better to give a more detailed block diagram or algorithm for a proposed method.
Author Response
Response to Reviewer 1 Comments
Point 1: The problem formulation is skipped. It simply narrates the previous work without mentioning the similarities and differences between this method and previous methods, and the innovation is not prominent
Response 1: I would like to express my gratitude to the referee for their valuable feedback. After carefully reviewing his/her suggestions, I found them to be very appropriate, particularly in terms of clarifying the problem definition. As a result, several changes have been made to the study. Article title has been changed to focus ensemble strategies for object detection. Thus, the ensemble strategies have been highlighted as the main focus of this work, and the relevant literature has been revisited to further support the problem definition.
Point 2: It is difficult for readers to understand the main contributions of this paper. The Introduction can be revised to emphasize the main contribution of the work
Response 2: Thank you for your valuable feedback. As suggested by the reviewer, the introduction section has been extensively revised in order to better highlight the contribution of our study. I reviewed 15 ensemble studies conducted on brain MRI datasets with regards to their objectives, methods, ensemble strategies, learning tasks, and time to publication. These studies were compared with the proposed study to better demonstrate its contribution. The following text and table were added to the article to expand the introduction section. The contribution was presented more clearly by comparing it with relevant literature.
Ensemble techniques are one of the methods used to improve performance in brain MRI studies for several tasks as in many other fields. According to the current ensemble studies summarized in Table 1, model ensemble can be achieved through two differ ent techniques, namely feature-level ensemble and combining model outputs. In the feature-level ensemble technique, the features of various inputs are combined during model training, while in the model output combining technique, the outputs of multiple models are combined using various strategies without being limited to a single classifier/segmentor/regressor model, resulting in more accurate and reliable results. As seen in Table 1, these studies are used in segmentation, classification, and regression tasks in many areas such as age estimation, Alzheimer’s detection, tumor segmentation, and tumor classification.
In Aurna et al. study , a two-stage ensemble based model is proposed for tumor classification problem by combining features extracted from deep learning architectures such as Custom CNN, EfficientNet-B0, and ResNet-50, and classifying them using classical machine learning methods such as Support Vector Machine (SVM), Random Forest (RF), etc. [22] . Similarly, Aamir et al. also combines features extracted from EfficientNet and ResNet50 architectures for ensemble learning to perform tumor classification. Feature-level ensemble strategies are also used for tumor segmentation task [23 ] . In Liu et al. study, an architecture called PIF-Net is proposed, which is based on the combination of features from different MRI modalities [ 24]. In Kua et al. study, brain age estimation with the scope of regression task is performed using ridge regression and support vector regression (SVR) with ResNet [25].
In contrast to feature-level ensemble strategies, Dolz et al. improves brain tissue segmentation performance for infants by combining the results of customized 3D CNN variants using the majority voting method [26 ]. For tumor segmentation task, Cabria et al. improves the performance by combining the results of Potential Field Segmentation, FOR, and PFC methods with rule-based [ 27 ], Feng et al. takes the average of the results of 3D Unet variants [28 ], and Das et al. combines the results of Basic Encoder-Decoder, U-Net, and SegNet models according to their success rates [29 ]. For the classification task, Tandel et al. combines the outputs of AlexNet, VGG16, ResNet18, GoogleNet, and ResNet50 models using the majority voting method [30 ], while study Islam et al. combines the outputs of DenseNet121, VGG19, and Inception V3 models [ 31 ] and Ghafourian et al. combines the outputs of SVM, Naive Bayes, and KNN [32] in a similar manner . In Kang et al. study , the results of multiple models are combined by taking their average values, resulting in improved tumor classification results [33 ]. For Parkinson’s detection, Kurmi et
- uses a fuzzy logic-based ensemble strategy with VGG16, ResNet50, Inception-V3, and Xception models [34 ], while for Alzheimer’s detection, Chatter et al. combines the outputs of SVM, logistic regression, naive Bayes, and K nearest neighbor methods using majority voting [35 ]. Finally, in Zahoor et al. study, both feature-level and model-level ensemble strategies are employed using multiple models for tumor classification [36] . Upon reviewing the literature, it has been observed that most of the ensemble studies have been conducted for the tasks of tumor classification and segmentation. To the best of the author’s knowledge, there is currently no comprehensive ensemble study for the object detection task for brain MRI studies. The main contributions of this paper are as follows:
- A comprehensive ensemble based object detection study for anatomical and pathological object detection in brain MRI .
- A total of nine state-of-the-art object detection models were employed to propose and evaluate four distinct ensemble strategies aimed at improving the accuracy and robustness of detecting anatomical and pathological regions in brain MRIs. The efficacy of these strategies was empirically assessed through rigorous experiments.
- A comparative evaluation of the current state-of-the-art object detection models for identifying anatomical and pathological regions in brain MRIs has been conducted as a benchmarking study in novel Gazi Brains 2020 dataset.
- Five different anatomical structures such as the brain tissue, eyes, optic nerves, lateral ventricles, third ventricle, and also pathological object as whole tumor parts seen in brain MRI were detected simultaneously.
Table 1. Current studies in brain MRI literature using ensemble strategies
|
Ref. |
Purpose |
Methods |
Ensemble Strategy |
Learning Task |
Year |
|
[26] |
Brain tissue segmentation for infant |
Custom 3D CNN |
Model outputs are combined using majority voting |
Segmentation |
2019 |
|
[27] |
Tumor segmentation |
Potential Field Segmentation, FOR and PFC |
Model outputs are combined by rule |
Segmentation |
2016 |
|
[30] |
Tumor classification |
AlexNet, VGG, ResNet, GoogleNet |
Model outputs are combined using majority voting |
Classification |
2022 |
|
[22] |
Tumor classification |
Custom CNN, EfficientNet-B0 and ResNet,Support Vector Machine,Random Forest, K Nearest Neigbour and AdaBoost. |
Feature level ensemble |
Classification |
2022 |
|
[23] |
Tumor classification |
EfficientNet and ResNet |
Feature level ensemble |
Classification |
2022 |
|
[35] |
Alzheimer disease classification |
SVM, logistic regression, naive Bayes, and K nearest neighbor |
Model outputs are combined using majority voting |
Classification |
2022 |
|
[34] |
Parkinson detection |
VGG16, ResNet, Inception-V3, and Xception |
Model outputs are combined using Fuzzy logic |
Classification |
2022 |
|
[28] |
Tumor segmentation and survival prediction |
3D U-Nets variants |
Model outputs are averaged |
Segmentation |
2020 |
|
[29] |
Tumor segmentation |
Basic Encoder-Decoder, U-Net and SegNet model |
Models outputs are combined based on accuracy |
Segmentation |
2022 |
|
[24] |
Tumor segmentation |
PIF-Net |
Pixel and Feature level ensemble |
Segmentation |
2022 |
|
[25] |
Age estimation |
ridge regression and support vector regression (SVR), Resnet |
Feature level ensemble |
Regression |
2021 |
|
[31] |
Tumor detection with federated learning |
DenseNet, VGG, and Inception V3 |
Model outputs are combined using majority voting |
Classification |
2022 |
|
[33] |
Tumor classification |
ResNet, DenseNet, VGG, AlexNet, Inception V3, |
Model outputs are averaged |
Classification |
2021 |
|
[36] |
Tumor classification |
VGG, SqueezeNet, GoogleNet, ResNet, XceptionNet,InceptionV3, ShuffleNet,DenseNet,SVM, MLP, and AdaBoost |
Features and |
Classification |
2022 |
|
[32] |
Tumor classification |
SVM, Naive Bayes, and KNN |
Model outputs are combined using majority voting |
Classification |
2023 |
|
This study |
Anathomical and Pathological Object Detection |
RetinaNet, YOLOv3, FCOS, NAS-FPN, ATSS, VFNet, Faster R-CNN, Dynamic R-CNN, Cascade R- CNN |
Model Bounding Box Outputs are recalculated using weighted sum |
Object Detection |
Point 3: The article has poor math formulating and a weak technical description.
Response 3: Thank you for your valuable feedback .The technical details of the methods used in the scope of the study are presented under the heading "2.2. Deep Learning Architectures for Anatomical and Pathological Object Detection". The mathematical functions that distinguish the models from each other are presented along with them. The added text can be found below.
Two-stage object detection models consist of two stages: Region Proposal Network (RPN) and Detector. In the RPN stage, candidate regions are proposed, and in the Detector stage, bounding box regression and classification are performed. Two-stage object detection models are also known as the R-CNN family, and there are many examples of this type of model. Faster R-CNN is an improved version of the Fast R-CNN model that has a faster run time. To realize this, it uses a CNN-based feature extractor for proposing rectangular objects instead of selective search in the proposal stage.the proposed object features are shared with the detector and used for bounding box regression and classification. A bounding box is defined as four different coordinates b = bx, by, bw, bh , and the regression is performed using the smoothed L1 loss function (Equation 2) between the ground truth bounding box and the candidate bounding box. The loss function learns the coordinates from the data by trying to minimize the distance. For the classification process, the classifier performs learning by minimizing the classification cross-entropy loss (Equation 1 for binary cross entropy) on the training set. During the learning process, positive and negative detections are identified based on the IoU metric according to the equation given in Equation 3
p stands for class probability and y is for ground truth label.
x is for regression label
b is for bounding box, G is for ground truth, T+ and T− stand for positive and negative
threshold for IoU respectively.
Cascade R-CNN performs object detection by taking into account different IoU overlap levels compared to Faster R-CNN. Cascade R-CNN architecture proposes a series of customized regressors for different IoU overlap threshold values, as given in Equation 4. N represents the total number of cascades stages and d denotes for sample distribution in equation.
Dynamic R-CNN involves a dynamic training procedure compared to Faster and Cascade R-CNN. In this method, instead of using predefined IoU threshold values in the second stage where regressor and classifier are used, a dynamically changing procedure based on the distribution of region proposals is proposed given in Equation 3. For this purpose, The Dynamic Label Assignment (DLA) process given in Equation 5 is first used according to the T threshold value updated based on the statistical distribution of proposals for object detection. For localization task (bounding box regression), Dynamic SmoothL1 Loss given in in Equation 6 is used. Similarly to DLA, DSL also changes regression labels based on the statistical distribution of proposals.
Tnow is for current IoU threshold and updates according to data distribution
βnow represents the hyper-parameters to control smooth loss.
The main difference between one-stage object detection models and two-stage object detection models is that one-stage models do not involve a region proposal stage. Therefore, they generally provide faster detector performance compared to two-stage methods. One- stage object detection models show diversity in many aspects. These models can be divided into various sub-categories based on the differences in their architectures (such as Anchor Based, Anchor Free, Feature Pyramid, Context, etc.). The following is a summary of the one-stage object detectors used in this study.
The most important component of the RetinaNet architecture is a loss function called Focal Loss given in Equation 7. This proposed loss function solves the problem of class imbalance during training. It allows the rarer class to be learned better than the classic cross-entropy loss (Equation 1), leading to a balanced model training. The Focal Loss equation is defined as Focal Loss. RetinaNet is an anchor-based architecture that uses Resnet-based FPN as its backbone. While the backbone is used to extract convolutional features from the input, two different subnets that take the backbone’s outputs as inputs are used for object classification and bounding box regression.
YoloV3 is a variant of Yolo, which is an improved version of YoloV2 in various aspects. The most important factor that makes Yolo models faster than other object detection models is that they do not contain a complex pipeline. In this architecture, the entire image is used as the input to the network, and bounding box regression and object detection are directly performed on this image. The main differences between YoloV3 and YoloV2 are that YoloV3 uses a deeper architecture called Darknet-53 for feature extraction and predicts bounding boxes in 3 different scales. FCOS is an anchor-free object detection model that performs object classification and bounding box regression without requiring overlap (IoU) calculations with anchors or detailed hyper-parameter tuning. Anchor-based methods consider the center of anchor boxes as the location in the input image and use it to regress the target bounding box. However, FCOS considers any point inside the ground truth box as a location and uses the distance t∗ = (l∗, t∗, r∗, b∗) (given in Equation 8) for regression. If a point is inside multiple bounding boxes, it is considered an ambiguous example, and multi-level prediction is used to reduce this situation. FCOS also uses the centerness strategy to improve bounding box detection quality (given in Equation 9). To do this, FCOS adds a parallel stage to the classification task to predict whether a location is a center or not. The centerness value is trained using binary cross-entropy loss and added to FCOS’s loss function.
l∗, t∗, r∗, b∗ are distances calculated from bounding box.
The NAS-FPN model has made improvements on the FPN model architecture used in object detection models for extracting visual representations. Unlike other models that use manual FPN architecture design, this model narrows down a wide search space for FPN architecture through the Neural Architecture Search algorithm to extract effective FPN architectures.
The ATSS model proposes a method that performs adaptive sampling of training set examples by utilizing the statistical characteristics of positive and negative examples, based on the fact that they have a significant impact on the model’s performance. This approach has led to significant improvements in both anchor-based and anchor-free detectors, filling the performance gap between these two different architectures.
VarifocalNet proposed a different approach compared to other models in evaluating candidate objects during training by considering not only a classification score or a combination of classification and localization scores but also an IoU-based classification score (IACS) that takes detection performance into account. This approach takes both localization accuracy and object confidence score into consideration, resulting in successful results, especially in dense object detection scenarios. To estimate the IACS score, Varifocal loss (given in Equation 10) and star-shaped bounding box representation were proposed.
p represents to predicted IACS score and q stands for target score
Point 4: The architecture of the framework needs to be described in more detail. The training process and how the parameters are set must be deepened.
Response 4: Thank you for your valuable feedback . As stated in the previous answer, technical details of the used architectures were provided. The hyperparameters used during model training are listed in Table 3. The ensemble strategies used in this study are also included under the "Experimental Setup" section. Whenever possible, default parameters (as specified in the mmdetection repository) were used in model training. This was done to avoid the need for an extensive hyperparameter optimization, and to demonstrate the effectiveness of the ensemble strategies in this study. Additionally, since 10-fold cross-validation was used, a total of 90 models were developed from 9 different models. As anticipated, such a large number of model trainings would be very computational cost if an extensive hyperparameter optimization were to be performed. Nonetheless, training with default parameters produced promising results as stated in Table 4,5,6.
Point 5: I would suggest the authors present more graphical information. It will be better to give a more detailed block diagram or algorithm for a proposed method.
Response 5: Thank you for your valuable feedback. As you suggested, a workflow diagram has been added to the article. Existing figures have been merged, and the process from data acquisition to the implementation of ensemble strategies has been added to the relevant figure. You can find the workflow image below.

Reviewer 2 Report
This study aims to combine decisions from multiple models using different ensemble strategies and to improve anatomical object detection model performance. In this study, nine different models were compared for the anatomical object detection problem and four different ensemble strategies were proposed. The proposed ensemble strategies ensured that each model individually boosted the anatomical object detection capacity and achieved the highest success rates by combining different models.
Strengths:
This paper is easy to follow. The idea to visualize the intermediate process of medical image segmentation is interesting.
The experimental results are promising.
Weakness:
How does the current work different from state-of-the-art? what is the unique research contribution.
What does it add to the subject area compared with other published material?
What specific improvements should the authors consider regarding the methodology? What further controls should be considered?
Ablation study is needed for verifying the proposed method.
The significance of this paper is not expounded sufficiently. The author needs to highlight this paper's innovative contributions.
The authors should include some promising paper in the LR: Roy, Sudipta, and Samir Kumar Bandyopadhyay. "A new method of brain tissues segmentation from MRI with accuracy estimation." Procedia Computer Science 85 (2016): 362-369. // Roy, Sudipta, Debnath Bhattacharyya, Samir Kumar Bandyopadhyay, and Tai-Hoon Kim. "An iterative implementation of level set for precise segmentation of brain tissues and abnormality detection from MR images." IETE Journal of Research 63, no. 6 (2017): 769-783. // Roy, Sudipta, Tanushree Meena, and Se-Jung Lim. "Demystifying supervised learning in healthcare 4.0: A new reality of transforming diagnostic medicine." Diagnostics 12, no. 10 (2022): 2549.
Please discuss the possible over-fitting risk by applying data growth study.
Author Response
Response to Reviewer 2 Comments
Point 1: How does the current work different from state-of-the-art? what is the unique research contribution
Response 1: I would like to express my gratitude to the referee for their valuable feedback, Several changes have been made to the study. Article title has been changed to focus ensemble strategies for object detection. Thus, the ensemble strategies have been highlighted as the main focus of this work, and the relevant literature has been revisited to further support the problem definition.
Point 2: What does it add to the subject area compared with other published material?
Response 2: Thank you for your valuable feedback. the introduction section has been extensively revised in order to better highlight the contribution of our study. I reviewed 15 ensemble studies conducted on brain MRI datasets with regards to their objectives, methods, ensemble strategies, learning tasks, and time to publication. These studies were compared with the proposed study to better demonstrate its contribution. The following text and table were added to the article to expand the introduction section. The contribution was presented more clearly by comparing it with relevant literature.
Ensemble techniques are one of the methods used to improve performance in brain MRI studies for several tasks as in many other fields. According to the current ensemble studies summarized in Table 1, model ensemble can be achieved through two differ ent techniques, namely feature-level ensemble and combining model outputs. In the feature-level ensemble technique, the features of various inputs are combined during model training, while in the model output combining technique, the outputs of multiple models are combined using various strategies without being limited to a single classifier/segmentor/regressor model, resulting in more accurate and reliable results. As seen in Table 1, these studies are used in segmentation, classification, and regression tasks in many areas such as age estimation, Alzheimer’s detection, tumor segmentation, and tumor classification.
In Aurna et al. study , a two-stage ensemble based model is proposed for tumor classification problem by combining features extracted from deep learning architectures such as Custom CNN, EfficientNet-B0, and ResNet-50, and classifying them using classical machine learning methods such as Support Vector Machine (SVM), Random Forest (RF), etc. [22] . Similarly, Aamir et al. also combines features extracted from EfficientNet and ResNet50 architectures for ensemble learning to perform tumor classification. Feature-level ensemble strategies are also used for tumor segmentation task [23 ] . In Liu et al. study, an architecture called PIF-Net is proposed, which is based on the combination of features from different MRI modalities [ 24]. In Kua et al. study, brain age estimation with the scope of regression task is performed using ridge regression and support vector regression (SVR) with ResNet [25].
In contrast to feature-level ensemble strategies, Dolz et al. improves brain tissue segmentation performance for infants by combining the results of customized 3D CNN variants using the majority voting method [26 ]. For tumor segmentation task, Cabria et al. improves the performance by combining the results of Potential Field Segmentation, FOR, and PFC methods with rule-based [ 27 ], Feng et al. takes the average of the results of 3D Unet variants [28 ], and Das et al. combines the results of Basic Encoder-Decoder, U-Net, and SegNet models according to their success rates [29 ]. For the classification task, Tandel et al. combines the outputs of AlexNet, VGG16, ResNet18, GoogleNet, and ResNet50 models using the majority voting method [30 ], while study Islam et al. combines the outputs of DenseNet121, VGG19, and Inception V3 models [ 31 ] and Ghafourian et al. combines the outputs of SVM, Naive Bayes, and KNN [32] in a similar manner . In Kang et al. study , the results of multiple models are combined by taking their average values, resulting in improved tumor classification results [33 ]. For Parkinson’s detection, Kurmi et
- uses a fuzzy logic-based ensemble strategy with VGG16, ResNet50, Inception-V3, and Xception models [34 ], while for Alzheimer’s detection, Chatter et al. combines the outputs of SVM, logistic regression, naive Bayes, and K nearest neighbor methods using majority voting [35 ]. Finally, in Zahoor et al. study, both feature-level and model-level ensemble strategies are employed using multiple models for tumor classification [36] . Upon reviewing the literature, it has been observed that most of the ensemble studies have been conducted for the tasks of tumor classification and segmentation. To the best of the author’s knowledge, there is currently no comprehensive ensemble study for the object detection task for brain MRI studies. The main contributions of this paper are as follows:
- A comprehensive ensemble based object detection study for anatomical and pathological object detection in brain MRI .
- A total of nine state-of-the-art object detection models were employed to propose and evaluate four distinct ensemble strategies aimed at improving the accuracy and robustness of detecting anatomical and pathological regions in brain MRIs. The efficacy of these strategies was empirically assessed through rigorous experiments.
- A comparative evaluation of the current state-of-the-art object detection models for identifying anatomical and pathological regions in brain MRIs has been conducted as a benchmarking study in novel Gazi Brains 2020 dataset.
- Five different anatomical structures such as the brain tissue, eyes, optic nerves, lateral ventricles, third ventricle, and also pathological object as whole tumor parts seen in brain MRI were detected simultaneously.
Table 1. Current studies in brain MRI literature using ensemble strategies
|
Ref. |
Purpose |
Methods |
Ensemble Strategy |
Learning Task |
Year |
|
[26] |
Brain tissue segmentation for infant |
Custom 3D CNN |
Model outputs are combined using majority voting |
Segmentation |
2019 |
|
[27] |
Tumor segmentation |
Potential Field Segmentation, FOR and PFC |
Model outputs are combined by rule |
Segmentation |
2016 |
|
[30] |
Tumor classification |
AlexNet, VGG, ResNet, GoogleNet |
Model outputs are combined using majority voting |
Classification |
2022 |
|
[22] |
Tumor classification |
Custom CNN, EfficientNet-B0 and ResNet,Support Vector Machine,Random Forest, K Nearest Neigbour and AdaBoost. |
Feature level ensemble |
Classification |
2022 |
|
[23] |
Tumor classification |
EfficientNet and ResNet |
Feature level ensemble |
Classification |
2022 |
|
[35] |
Alzheimer disease classification |
SVM, logistic regression, naive Bayes, and K nearest neighbor |
Model outputs are combined using majority voting |
Classification |
2022 |
|
[34] |
Parkinson detection |
VGG16, ResNet, Inception-V3, and Xception |
Model outputs are combined using Fuzzy logic |
Classification |
2022 |
|
[28] |
Tumor segmentation and survival prediction |
3D U-Nets variants |
Model outputs are averaged |
Segmentation |
2020 |
|
[29] |
Tumor segmentation |
Basic Encoder-Decoder, U-Net and SegNet model |
Models outputs are combined based on accuracy |
Segmentation |
2022 |
|
[24] |
Tumor segmentation |
PIF-Net |
Pixel and Feature level ensemble |
Segmentation |
2022 |
|
[25] |
Age estimation |
ridge regression and support vector regression (SVR), Resnet |
Feature level ensemble |
Regression |
2021 |
|
[31] |
Tumor detection with federated learning |
DenseNet, VGG, and Inception V3 |
Model outputs are combined using majority voting |
Classification |
2022 |
|
[33] |
Tumor classification |
ResNet, DenseNet, VGG, AlexNet, Inception V3, |
Model outputs are averaged |
Classification |
2021 |
|
[36] |
Tumor classification |
VGG, SqueezeNet, GoogleNet, ResNet, XceptionNet,InceptionV3, ShuffleNet,DenseNet,SVM, MLP, and AdaBoost |
Features and |
Classification |
2022 |
|
[32] |
Tumor classification |
SVM, Naive Bayes, and KNN |
Model outputs are combined using majority voting |
Classification |
2023 |
|
This study |
Anathomical and Pathological Object Detection |
RetinaNet, YOLOv3, FCOS, NAS-FPN, ATSS, VFNet, Faster R-CNN, Dynamic R-CNN, Cascade R- CNN |
Model Bounding Box Outputs are recalculated using weighted sum |
Object Detection |
Point 3: What specific improvements should the authors consider regarding the methodology? What further controls should be considered?.
Response 3: As shown in Table 1 (as state in previous answer), in this study, for the first time, ensemble strategies of object detection models have been applied on brain MRI data, and their effectiveness has been demonstrated. I have also expanded methodology section by giving technical details of used architectures. Added text can be seen as below.
The technical details of the methods used in the scope of the study are presented under the heading "2.2. Deep Learning Architectures for Anatomical and Pathological Object Detection". The mathematical functions that distinguish the models from each other are presented along with them. The added text can be found below.
Two-stage object detection models consist of two stages: Region Proposal Network (RPN) and Detector. In the RPN stage, candidate regions are proposed, and in the Detector stage, bounding box regression and classification are performed. Two-stage object detection models are also known as the R-CNN family, and there are many examples of this type of model. Faster R-CNN is an improved version of the Fast R-CNN model that has a faster run time. To realize this, it uses a CNN-based feature extractor for proposing rectangular objects instead of selective search in the proposal stage.the proposed object features are shared with the detector and used for bounding box regression and classification. A bounding box is defined as four different coordinates b = bx, by, bw, bh , and the regression is performed using the smoothed L1 loss function (Equation 2) between the ground truth bounding box and the candidate bounding box. The loss function learns the coordinates from the data by trying to minimize the distance. For the classification process, the classifier performs learning by minimizing the classification cross-entropy loss (Equation 1 for binary cross entropy) on the training set. During the learning process, positive and negative detections are identified based on the IoU metric according to the equation given in Equation 3
p stands for class probability and y is for ground truth label.
x is for regression label
b is for bounding box, G is for ground truth, T+ and T− stand for positive and negative
threshold for IoU respectively.
Cascade R-CNN performs object detection by taking into account different IoU overlap levels compared to Faster R-CNN. Cascade R-CNN architecture proposes a series of customized regressors for different IoU overlap threshold values, as given in Equation 4. N represents the total number of cascades stages and d denotes for sample distribution in equation.
Dynamic R-CNN involves a dynamic training procedure compared to Faster and Cascade R-CNN. In this method, instead of using predefined IoU threshold values in the second stage where regressor and classifier are used, a dynamically changing procedure based on the distribution of region proposals is proposed given in Equation 3. For this purpose, The Dynamic Label Assignment (DLA) process given in Equation 5 is first used according to the T threshold value updated based on the statistical distribution of proposals for object detection. For localization task (bounding box regression), Dynamic SmoothL1 Loss given in in Equation 6 is used. Similarly to DLA, DSL also changes regression labels based on the statistical distribution of proposals.
Tnow is for current IoU threshold and updates according to data distribution
βnow represents the hyper-parameters to control smooth loss.
The main difference between one-stage object detection models and two-stage object detection models is that one-stage models do not involve a region proposal stage. Therefore, they generally provide faster detector performance compared to two-stage methods. One- stage object detection models show diversity in many aspects. These models can be divided into various sub-categories based on the differences in their architectures (such as Anchor Based, Anchor Free, Feature Pyramid, Context, etc.). The following is a summary of the one-stage object detectors used in this study.
The most important component of the RetinaNet architecture is a loss function called Focal Loss given in Equation 7. This proposed loss function solves the problem of class imbalance during training. It allows the rarer class to be learned better than the classic cross-entropy loss (Equation 1), leading to a balanced model training. The Focal Loss equation is defined as Focal Loss. RetinaNet is an anchor-based architecture that uses Resnet-based FPN as its backbone. While the backbone is used to extract convolutional features from the input, two different subnets that take the backbone’s outputs as inputs are used for object classification and bounding box regression.
YoloV3 is a variant of Yolo, which is an improved version of YoloV2 in various aspects. The most important factor that makes Yolo models faster than other object detection models is that they do not contain a complex pipeline. In this architecture, the entire image is used as the input to the network, and bounding box regression and object detection are directly performed on this image. The main differences between YoloV3 and YoloV2 are that YoloV3 uses a deeper architecture called Darknet-53 for feature extraction and predicts bounding boxes in 3 different scales. FCOS is an anchor-free object detection model that performs object classification and bounding box regression without requiring overlap (IoU) calculations with anchors or detailed hyper-parameter tuning. Anchor-based methods consider the center of anchor boxes as the location in the input image and use it to regress the target bounding box. However, FCOS considers any point inside the ground truth box as a location and uses the distance t∗ = (l∗, t∗, r∗, b∗) (given in Equation 8) for regression. If a point is inside multiple bounding boxes, it is considered an ambiguous example, and multi-level prediction is used to reduce this situation. FCOS also uses the centerness strategy to improve bounding box detection quality (given in Equation 9). To do this, FCOS adds a parallel stage to the classification task to predict whether a location is a center or not. The centerness value is trained using binary cross-entropy loss and added to FCOS’s loss function.
l∗, t∗, r∗, b∗ are distances calculated from bounding box.
The NAS-FPN model has made improvements on the FPN model architecture used in object detection models for extracting visual representations. Unlike other models that use manual FPN architecture design, this model narrows down a wide search space for FPN architecture through the Neural Architecture Search algorithm to extract effective FPN architectures.
The ATSS model proposes a method that performs adaptive sampling of training set examples by utilizing the statistical characteristics of positive and negative examples, based on the fact that they have a significant impact on the model’s performance. This approach has led to significant improvements in both anchor-based and anchor-free detectors, filling the performance gap between these two different architectures.
VarifocalNet proposed a different approach compared to other models in evaluating candidate objects during training by considering not only a classification score or a combination of classification and localization scores but also an IoU-based classification score (IACS) that takes detection performance into account. This approach takes both localization accuracy and object confidence score into consideration, resulting in successful results, especially in dense object detection scenarios. To estimate the IACS score, Varifocal loss (given in Equation 10) and star-shaped bounding box representation were proposed.
p represents to predicted IACS score and q stands for target score
Point 4: Ablation study is needed for verifying the proposed method.
Response 4: Thank you for your valuable feedback . In this study, four different ensemble strategies were proposed and their effectiveness was measured. If the effect of hyperparameters of the base models used in the ablation study is desired, as deemed appropriate by the reviewer, a total of 90 models were developed using 10-fold cross-validation for 9 architectures. With such intensive computational costs, hyperparameters were used as default settings as much as possible. Thus, the efficiency and use case of the ensemble strategies were demonstrated without intensive hyper parameter optimization. Nonetheless, training with default parameters produced promising results as stated in Table 4,5,6.
Point 5: The significance of this paper is not expounded sufficiently. The author needs to highlight this paper's innovative contributions
Response 5: Thank you for your valuable feedback. the introduction section has been extensively revised in order to better highlight the contribution of our study. I reviewed another 15 ensemble studies conducted on brain MRI datasets with regards to their objectives, methods, ensemble strategies, learning tasks, and time to publication. These studies were compared with the proposed study to better demonstrate its contribution. The contribution was presented more clearly by comparing it with relevant literature. . As a result, several changes have been made to the study. Article title has been changed to focus ensemble strategies for object detection. Thus, the ensemble strategies have been highlighted as the main focus of this work, and the relevant literature has been revisited to further support the problem definition.
Point 6: The authors should include some promising paper in the LR: Roy, Sudipta, and Samir Kumar Bandyopadhyay. "A new method of brain tissues segmentation from MRI with accuracy estimation." Procedia Computer Science 85 (2016): 362-369. // Roy, Sudipta, Debnath Bhattacharyya, Samir Kumar Bandyopadhyay, and Tai-Hoon Kim. "An iterative implementation of level set for precise segmentation of brain tissues and abnormality detection from MR images." IETE Journal of Research 63, no. 6 (2017): 769-783. // Roy, Sudipta, Tanushree Meena, and Se-Jung Lim. "Demystifying supervised learning in healthcare 4.0: A new reality of transforming diagnostic medicine." Diagnostics 12, no. 10 (2022): 2549.
Response 6: Thank you for reminding me of these valuable studies. I have cited them in the introduction section
Point 7: Please discuss the possible over-fitting risk by applying data growth study.
Response 7: Thanks for valuable feedback. In the discussion section, I have added text given in below.
As in many of the studies listed in Table 1, this study was also conducted with a single dataset. This can lead to overfitting risks for deep learning models developed on limited datasets, which can limit their generalization abilities. Although it may be possible to combine data from different open sources, finding datasets with similar labels or similar tasks is almost impossible. In this study, the success results of models developed with the 10-fold cross-validation method on the Gazi Brains open dataset were provided, and the generalization abilities within the dataset were measured for different combinations of data.

Round 2
Reviewer 1 Report
The changes skipped in the revised file.
Author Response
Response to Reviewer 1 Comments
Point 1: The problem formulation is skipped. It simply narrates the previous work without mentioning the similarities and differences between this method and previous methods, and the innovation is not prominent
Response 1: I would like to express my gratitude to the referee for their valuable feedback. After carefully reviewing his/her suggestions, I found them to be very appropriate, particularly in terms of clarifying the problem definition. As a result, several changes have been made to the study. Article title has been changed to focus ensemble strategies for object detection. Thus, the ensemble strategies have been highlighted as the main focus of this work, and the relevant literature has been revisited to further support the problem definition.
Point 2: It is difficult for readers to understand the main contributions of this paper. The Introduction can be revised to emphasize the main contribution of the work
Response 2: Thank you for your valuable feedback. As suggested by the reviewer, the introduction section has been extensively revised in order to better highlight the contribution of our study. I reviewed 15 ensemble studies conducted on brain MRI datasets with regards to their objectives, methods, ensemble strategies, learning tasks, and time to publication. These studies were compared with the proposed study to better demonstrate its contribution. The following text and table were added to the article to expand the introduction section. The contribution was presented more clearly by comparing it with relevant literature.
Ensemble techniques are one of the methods used to improve performance in brain MRI studies for several tasks as in many other fields. According to the current ensemble studies summarized in Table 1, model ensemble can be achieved through two differ ent techniques, namely feature-level ensemble and combining model outputs. In the feature-level ensemble technique, the features of various inputs are combined during model training, while in the model output combining technique, the outputs of multiple models are combined using various strategies without being limited to a single classifier/segmentor/regressor model, resulting in more accurate and reliable results. As seen in Table 1, these studies are used in segmentation, classification, and regression tasks in many areas such as age estimation, Alzheimer’s detection, tumor segmentation, and tumor classification.
In Aurna et al. study , a two-stage ensemble based model is proposed for tumor classification problem by combining features extracted from deep learning architectures such as Custom CNN, EfficientNet-B0, and ResNet-50, and classifying them using classical machine learning methods such as Support Vector Machine (SVM), Random Forest (RF), etc. [22] . Similarly, Aamir et al. also combines features extracted from EfficientNet and ResNet50 architectures for ensemble learning to perform tumor classification. Feature-level ensemble strategies are also used for tumor segmentation task [23 ] . In Liu et al. study, an architecture called PIF-Net is proposed, which is based on the combination of features from different MRI modalities [ 24]. In Kua et al. study, brain age estimation with the scope of regression task is performed using ridge regression and support vector regression (SVR) with ResNet [25].
In contrast to feature-level ensemble strategies, Dolz et al. improves brain tissue segmentation performance for infants by combining the results of customized 3D CNN variants using the majority voting method [26 ]. For tumor segmentation task, Cabria et al. improves the performance by combining the results of Potential Field Segmentation, FOR, and PFC methods with rule-based [ 27 ], Feng et al. takes the average of the results of 3D Unet variants [28 ], and Das et al. combines the results of Basic Encoder-Decoder, U-Net, and SegNet models according to their success rates [29 ]. For the classification task, Tandel et al. combines the outputs of AlexNet, VGG16, ResNet18, GoogleNet, and ResNet50 models using the majority voting method [30 ], while study Islam et al. combines the outputs of DenseNet121, VGG19, and Inception V3 models [ 31 ] and Ghafourian et al. combines the outputs of SVM, Naive Bayes, and KNN [32] in a similar manner . In Kang et al. study , the results of multiple models are combined by taking their average values, resulting in improved tumor classification results [33 ]. For Parkinson’s detection, Kurmi et
- uses a fuzzy logic-based ensemble strategy with VGG16, ResNet50, Inception-V3, and Xception models [34 ], while for Alzheimer’s detection, Chatter et al. combines the outputs of SVM, logistic regression, naive Bayes, and K nearest neighbor methods using majority voting [35 ]. Finally, in Zahoor et al. study, both feature-level and model-level ensemble strategies are employed using multiple models for tumor classification [36] . Upon reviewing the literature, it has been observed that most of the ensemble studies have been conducted for the tasks of tumor classification and segmentation. To the best of the author’s knowledge, there is currently no comprehensive ensemble study for the object detection task for brain MRI studies. The main contributions of this paper are as follows:
- A comprehensive ensemble based object detection study for anatomical and pathological object detection in brain MRI .
- A total of nine state-of-the-art object detection models were employed to propose and evaluate four distinct ensemble strategies aimed at improving the accuracy and robustness of detecting anatomical and pathological regions in brain MRIs. The efficacy of these strategies was empirically assessed through rigorous experiments.
- A comparative evaluation of the current state-of-the-art object detection models for identifying anatomical and pathological regions in brain MRIs has been conducted as a benchmarking study in novel Gazi Brains 2020 dataset.
- Five different anatomical structures such as the brain tissue, eyes, optic nerves, lateral ventricles, third ventricle, and also pathological object as whole tumor parts seen in brain MRI were detected simultaneously.
Table 1. Current studies in brain MRI literature using ensemble strategies
|
Ref. |
Purpose |
Methods |
Ensemble Strategy |
Learning Task |
Year |
|
[26] |
Brain tissue segmentation for infant |
Custom 3D CNN |
Model outputs are combined using majority voting |
Segmentation |
2019 |
|
[27] |
Tumor segmentation |
Potential Field Segmentation, FOR and PFC |
Model outputs are combined by rule |
Segmentation |
2016 |
|
[30] |
Tumor classification |
AlexNet, VGG, ResNet, GoogleNet |
Model outputs are combined using majority voting |
Classification |
2022 |
|
[22] |
Tumor classification |
Custom CNN, EfficientNet-B0 and ResNet,Support Vector Machine,Random Forest, K Nearest Neigbour and AdaBoost. |
Feature level ensemble |
Classification |
2022 |
|
[23] |
Tumor classification |
EfficientNet and ResNet |
Feature level ensemble |
Classification |
2022 |
|
[35] |
Alzheimer disease classification |
SVM, logistic regression, naive Bayes, and K nearest neighbor |
Model outputs are combined using majority voting |
Classification |
2022 |
|
[34] |
Parkinson detection |
VGG16, ResNet, Inception-V3, and Xception |
Model outputs are combined using Fuzzy logic |
Classification |
2022 |
|
[28] |
Tumor segmentation and survival prediction |
3D U-Nets variants |
Model outputs are averaged |
Segmentation |
2020 |
|
[29] |
Tumor segmentation |
Basic Encoder-Decoder, U-Net and SegNet model |
Models outputs are combined based on accuracy |
Segmentation |
2022 |
|
[24] |
Tumor segmentation |
PIF-Net |
Pixel and Feature level ensemble |
Segmentation |
2022 |
|
[25] |
Age estimation |
ridge regression and support vector regression (SVR), Resnet |
Feature level ensemble |
Regression |
2021 |
|
[31] |
Tumor detection with federated learning |
DenseNet, VGG, and Inception V3 |
Model outputs are combined using majority voting |
Classification |
2022 |
|
[33] |
Tumor classification |
ResNet, DenseNet, VGG, AlexNet, Inception V3, |
Model outputs are averaged |
Classification |
2021 |
|
[36] |
Tumor classification |
VGG, SqueezeNet, GoogleNet, ResNet, XceptionNet,InceptionV3, ShuffleNet,DenseNet,SVM, MLP, and AdaBoost |
Features and |
Classification |
2022 |
|
[32] |
Tumor classification |
SVM, Naive Bayes, and KNN |
Model outputs are combined using majority voting |
Classification |
2023 |
|
This study |
Anathomical and Pathological Object Detection |
RetinaNet, YOLOv3, FCOS, NAS-FPN, ATSS, VFNet, Faster R-CNN, Dynamic R-CNN, Cascade R- CNN |
Model Bounding Box Outputs are recalculated using weighted sum |
Object Detection |
Point 3: The article has poor math formulating and a weak technical description.
Response 3: Thank you for your valuable feedback .The technical details of the methods used in the scope of the study are presented under the heading "2.2. Deep Learning Architectures for Anatomical and Pathological Object Detection". The mathematical functions that distinguish the models from each other are presented along with them. In addition, a new experimental setup was designed. This allowed for FROC curves to be presented for each individual anatomical and pathological region, containing information on FFPI vs TPR at various confidence scores, for the best individual models and ensemble strategies. The FROC curves were evaluated using the FAUC score (the area under the curve). Furthermore, experiments were conducted on the frequently used BraTS 2020 HGG dataset with 9 different models, and the ensemble strategy was compared to the best models by providing FROC graphs. The added text and graphs can be found below.
Two-stage object detection models consist of two stages: Region Proposal Network (RPN) and Detector. In the RPN stage, candidate regions are proposed, and in the Detector stage, bounding box regression and classification are performed. Two-stage object detection models are also known as the R-CNN family, and there are many examples of this type of model. Faster R-CNN is an improved version of the Fast R-CNN model that has a faster run time. To realize this, it uses a CNN-based feature extractor for proposing rectangular objects instead of selective search in the proposal stage.the proposed object features are shared with the detector and used for bounding box regression and classification. A bounding box is defined as four different coordinates b = bx, by, bw, bh , and the regression is performed using the smoothed L1 loss function (Equation 2) between the ground truth bounding box and the candidate bounding box. The loss function learns the coordinates from the data by trying to minimize the distance. For the classification process, the classifier performs learning by minimizing the classification cross-entropy loss (Equation 1 for binary cross entropy) on the training set. During the learning process, positive and negative detections are identified based on the IoU metric according to the equation given in Equation 3
p stands for class probability and y is for ground truth label.
x is for regression label
b is for bounding box, G is for ground truth, T+ and T− stand for positive and negative
threshold for IoU respectively.
Cascade R-CNN performs object detection by taking into account different IoU overlap levels compared to Faster R-CNN. Cascade R-CNN architecture proposes a series of customized regressors for different IoU overlap threshold values, as given in Equation 4. N represents the total number of cascades stages and d denotes for sample distribution in equation.
Dynamic R-CNN involves a dynamic training procedure compared to Faster and Cascade R-CNN. In this method, instead of using predefined IoU threshold values in the second stage where regressor and classifier are used, a dynamically changing procedure based on the distribution of region proposals is proposed given in Equation 3. For this purpose, The Dynamic Label Assignment (DLA) process given in Equation 5 is first used according to the T threshold value updated based on the statistical distribution of proposals for object detection. For localization task (bounding box regression), Dynamic SmoothL1 Loss given in in Equation 6 is used. Similarly to DLA, DSL also changes regression labels based on the statistical distribution of proposals.
Tnow is for current IoU threshold and updates according to data distribution
βnow represents the hyper-parameters to control smooth loss.
The main difference between one-stage object detection models and two-stage object detection models is that one-stage models do not involve a region proposal stage. Therefore, they generally provide faster detector performance compared to two-stage methods. One- stage object detection models show diversity in many aspects. These models can be divided into various sub-categories based on the differences in their architectures (such as Anchor Based, Anchor Free, Feature Pyramid, Context, etc.). The following is a summary of the one-stage object detectors used in this study.
The most important component of the RetinaNet architecture is a loss function called Focal Loss given in Equation 7. This proposed loss function solves the problem of class imbalance during training. It allows the rarer class to be learned better than the classic cross-entropy loss (Equation 1), leading to a balanced model training. The Focal Loss equation is defined as Focal Loss. RetinaNet is an anchor-based architecture that uses Resnet-based FPN as its backbone. While the backbone is used to extract convolutional features from the input, two different subnets that take the backbone’s outputs as inputs are used for object classification and bounding box regression.
YoloV3 is a variant of Yolo, which is an improved version of YoloV2 in various aspects. The most important factor that makes Yolo models faster than other object detection models is that they do not contain a complex pipeline. In this architecture, the entire image is used as the input to the network, and bounding box regression and object detection are directly performed on this image. The main differences between YoloV3 and YoloV2 are that YoloV3 uses a deeper architecture called Darknet-53 for feature extraction and predicts bounding boxes in 3 different scales. FCOS is an anchor-free object detection model that performs object classification and bounding box regression without requiring overlap (IoU) calculations with anchors or detailed hyper-parameter tuning. Anchor-based methods consider the center of anchor boxes as the location in the input image and use it to regress the target bounding box. However, FCOS considers any point inside the ground truth box as a location and uses the distance t∗ = (l∗, t∗, r∗, b∗) (given in Equation 8) for regression. If a point is inside multiple bounding boxes, it is considered an ambiguous example, and multi-level prediction is used to reduce this situation. FCOS also uses the centerness strategy to improve bounding box detection quality (given in Equation 9). To do this, FCOS adds a parallel stage to the classification task to predict whether a location is a center or not. The centerness value is trained using binary cross-entropy loss and added to FCOS’s loss function.
l∗, t∗, r∗, b∗ are distances calculated from bounding box.
The NAS-FPN model has made improvements on the FPN model architecture used in object detection models for extracting visual representations. Unlike other models that use manual FPN architecture design, this model narrows down a wide search space for FPN architecture through the Neural Architecture Search algorithm to extract effective FPN architectures.
The ATSS model proposes a method that performs adaptive sampling of training set examples by utilizing the statistical characteristics of positive and negative examples, based on the fact that they have a significant impact on the model’s performance. This approach has led to significant improvements in both anchor-based and anchor-free detectors, filling the performance gap between these two different architectures.
VarifocalNet proposed a different approach compared to other models in evaluating candidate objects during training by considering not only a classification score or a combination of classification and localization scores but also an IoU-based classification score (IACS) that takes detection performance into account. This approach takes both localization accuracy and object confidence score into consideration, resulting in successful results, especially in dense object detection scenarios. To estimate the IACS score, Varifocal loss (given in Equation 10) and star-shaped bounding box representation were proposed.
p represents to predicted IACS score and q stands for target score
Figure 3. FROC curves based on TPR and FPPI for each anatomical and pathological region for the
best ensemble strategy and the best individual models on Gazi Brains 2020 datase
Figure 4. FROC curve for the pathological region for the best different model strategy on BraTS 2020
Dataset
Point 4: The architecture of the framework needs to be described in more detail. The training process and how the parameters are set must be deepened.
Response 4: Thank you for your valuable feedback . As stated in the previous answer, technical details of the used architectures were provided. The hyperparameters used during model training are listed in Table 3. The ensemble strategies used in this study are also included under the "Experimental Setup" section. Whenever possible, default parameters (as specified in the mmdetection repository) were used in model training. This was done to avoid the need for an extensive hyperparameter optimization, and to demonstrate the effectiveness of the ensemble strategies in this study. Additionally, since 10-fold cross-validation was used, a total of 90 models were developed from 9 different models. As anticipated, such a large number of model trainings would be very computational cost if an extensive hyperparameter optimization were to be performed. Nonetheless, training with default parameters produced promising results as stated in Table 4,5,6.
Point 5: I would suggest the authors present more graphical information. It will be better to give a more detailed block diagram or algorithm for a proposed method.
Response 5: Thank you for your valuable feedback. As you suggested, a workflow diagram has been added to the article. Existing figures have been merged, and the process from data acquisition to the implementation of ensemble strategies has been added to the relevant figure. You can find the workflow image below.

Reviewer 2 Report
Either author has missed to update the file or not updated. I don't see any changes on the revised file.
My reviews are same as the previous. I need highlighted response in the main manuscript.
Author Response
Response to Reviewer 2 Comments
Point 1: How does the current work different from state-of-the-art? what is the unique research contribution
Response 1: I would like to express my gratitude to the referee for their valuable feedback, Several changes have been made to the study. Article title has been changed to focus ensemble strategies for object detection. Thus, the ensemble strategies have been highlighted as the main focus of this work, and the relevant literature has been revisited to further support the problem definition.
Point 2: What does it add to the subject area compared with other published material?
Response 2: Thank you for your valuable feedback. the introduction section has been extensively revised in order to better highlight the contribution of our study. I reviewed 15 ensemble studies conducted on brain MRI datasets with regards to their objectives, methods, ensemble strategies, learning tasks, and time to publication. These studies were compared with the proposed study to better demonstrate its contribution. The following text and table were added to the article to expand the introduction section. The contribution was presented more clearly by comparing it with relevant literature.
Ensemble techniques are one of the methods used to improve performance in brain MRI studies for several tasks as in many other fields. According to the current ensemble studies summarized in Table 1, model ensemble can be achieved through two differ ent techniques, namely feature-level ensemble and combining model outputs. In the feature-level ensemble technique, the features of various inputs are combined during model training, while in the model output combining technique, the outputs of multiple models are combined using various strategies without being limited to a single classifier/segmentor/regressor model, resulting in more accurate and reliable results. As seen in Table 1, these studies are used in segmentation, classification, and regression tasks in many areas such as age estimation, Alzheimer’s detection, tumor segmentation, and tumor classification.
In Aurna et al. study , a two-stage ensemble based model is proposed for tumor classification problem by combining features extracted from deep learning architectures such as Custom CNN, EfficientNet-B0, and ResNet-50, and classifying them using classical machine learning methods such as Support Vector Machine (SVM), Random Forest (RF), etc. [22] . Similarly, Aamir et al. also combines features extracted from EfficientNet and ResNet50 architectures for ensemble learning to perform tumor classification. Feature-level ensemble strategies are also used for tumor segmentation task [23 ] . In Liu et al. study, an architecture called PIF-Net is proposed, which is based on the combination of features from different MRI modalities [ 24]. In Kua et al. study, brain age estimation with the scope of regression task is performed using ridge regression and support vector regression (SVR) with ResNet [25].
In contrast to feature-level ensemble strategies, Dolz et al. improves brain tissue segmentation performance for infants by combining the results of customized 3D CNN variants using the majority voting method [26 ]. For tumor segmentation task, Cabria et al. improves the performance by combining the results of Potential Field Segmentation, FOR, and PFC methods with rule-based [ 27 ], Feng et al. takes the average of the results of 3D Unet variants [28 ], and Das et al. combines the results of Basic Encoder-Decoder, U-Net, and SegNet models according to their success rates [29 ]. For the classification task, Tandel et al. combines the outputs of AlexNet, VGG16, ResNet18, GoogleNet, and ResNet50 models using the majority voting method [30 ], while study Islam et al. combines the outputs of DenseNet121, VGG19, and Inception V3 models [ 31 ] and Ghafourian et al. combines the outputs of SVM, Naive Bayes, and KNN [32] in a similar manner . In Kang et al. study , the results of multiple models are combined by taking their average values, resulting in improved tumor classification results [33 ]. For Parkinson’s detection, Kurmi et
- uses a fuzzy logic-based ensemble strategy with VGG16, ResNet50, Inception-V3, and Xception models [34 ], while for Alzheimer’s detection, Chatter et al. combines the outputs of SVM, logistic regression, naive Bayes, and K nearest neighbor methods using majority voting [35 ]. Finally, in Zahoor et al. study, both feature-level and model-level ensemble strategies are employed using multiple models for tumor classification [36] . Upon reviewing the literature, it has been observed that most of the ensemble studies have been conducted for the tasks of tumor classification and segmentation. To the best of the author’s knowledge, there is currently no comprehensive ensemble study for the object detection task for brain MRI studies. The main contributions of this paper are as follows:
- A comprehensive ensemble based object detection study for anatomical and pathological object detection in brain MRI .
- A total of nine state-of-the-art object detection models were employed to propose and evaluate four distinct ensemble strategies aimed at improving the accuracy and robustness of detecting anatomical and pathological regions in brain MRIs. The efficacy of these strategies was empirically assessed through rigorous experiments.
- A comparative evaluation of the current state-of-the-art object detection models for identifying anatomical and pathological regions in brain MRIs has been conducted as a benchmarking study in novel Gazi Brains 2020 dataset.
- Five different anatomical structures such as the brain tissue, eyes, optic nerves, lateral ventricles, third ventricle, and also pathological object as whole tumor parts seen in brain MRI were detected simultaneously.
Table 1. Current studies in brain MRI literature using ensemble strategies
|
Ref. |
Purpose |
Methods |
Ensemble Strategy |
Learning Task |
Year |
|
[26] |
Brain tissue segmentation for infant |
Custom 3D CNN |
Model outputs are combined using majority voting |
Segmentation |
2019 |
|
[27] |
Tumor segmentation |
Potential Field Segmentation, FOR and PFC |
Model outputs are combined by rule |
Segmentation |
2016 |
|
[30] |
Tumor classification |
AlexNet, VGG, ResNet, GoogleNet |
Model outputs are combined using majority voting |
Classification |
2022 |
|
[22] |
Tumor classification |
Custom CNN, EfficientNet-B0 and ResNet,Support Vector Machine,Random Forest, K Nearest Neigbour and AdaBoost. |
Feature level ensemble |
Classification |
2022 |
|
[23] |
Tumor classification |
EfficientNet and ResNet |
Feature level ensemble |
Classification |
2022 |
|
[35] |
Alzheimer disease classification |
SVM, logistic regression, naive Bayes, and K nearest neighbor |
Model outputs are combined using majority voting |
Classification |
2022 |
|
[34] |
Parkinson detection |
VGG16, ResNet, Inception-V3, and Xception |
Model outputs are combined using Fuzzy logic |
Classification |
2022 |
|
[28] |
Tumor segmentation and survival prediction |
3D U-Nets variants |
Model outputs are averaged |
Segmentation |
2020 |
|
[29] |
Tumor segmentation |
Basic Encoder-Decoder, U-Net and SegNet model |
Models outputs are combined based on accuracy |
Segmentation |
2022 |
|
[24] |
Tumor segmentation |
PIF-Net |
Pixel and Feature level ensemble |
Segmentation |
2022 |
|
[25] |
Age estimation |
ridge regression and support vector regression (SVR), Resnet |
Feature level ensemble |
Regression |
2021 |
|
[31] |
Tumor detection with federated learning |
DenseNet, VGG, and Inception V3 |
Model outputs are combined using majority voting |
Classification |
2022 |
|
[33] |
Tumor classification |
ResNet, DenseNet, VGG, AlexNet, Inception V3, |
Model outputs are averaged |
Classification |
2021 |
|
[36] |
Tumor classification |
VGG, SqueezeNet, GoogleNet, ResNet, XceptionNet,InceptionV3, ShuffleNet,DenseNet,SVM, MLP, and AdaBoost |
Features and |
Classification |
2022 |
|
[32] |
Tumor classification |
SVM, Naive Bayes, and KNN |
Model outputs are combined using majority voting |
Classification |
2023 |
|
This study |
Anathomical and Pathological Object Detection |
RetinaNet, YOLOv3, FCOS, NAS-FPN, ATSS, VFNet, Faster R-CNN, Dynamic R-CNN, Cascade R- CNN |
Model Bounding Box Outputs are recalculated using weighted sum |
Object Detection |
Point 3: What specific improvements should the authors consider regarding the methodology? What further controls should be considered?.
Response 3: As shown in Table 1 (as state in previous answer), in this study, for the first time, ensemble strategies of object detection models have been applied on brain MRI data, and their effectiveness has been demonstrated. I have also expanded methodology section by giving technical details of used architectures. Added text can be seen as below.
The technical details of the methods used in the scope of the study are presented under the heading "2.2. Deep Learning Architectures for Anatomical and Pathological Object Detection". The mathematical functions that distinguish the models from each other are presented along with them. The added text can be found below.
Two-stage object detection models consist of two stages: Region Proposal Network (RPN) and Detector. In the RPN stage, candidate regions are proposed, and in the Detector stage, bounding box regression and classification are performed. Two-stage object detection models are also known as the R-CNN family, and there are many examples of this type of model. Faster R-CNN is an improved version of the Fast R-CNN model that has a faster run time. To realize this, it uses a CNN-based feature extractor for proposing rectangular objects instead of selective search in the proposal stage.the proposed object features are shared with the detector and used for bounding box regression and classification. A bounding box is defined as four different coordinates b = bx, by, bw, bh , and the regression is performed using the smoothed L1 loss function (Equation 2) between the ground truth bounding box and the candidate bounding box. The loss function learns the coordinates from the data by trying to minimize the distance. For the classification process, the classifier performs learning by minimizing the classification cross-entropy loss (Equation 1 for binary cross entropy) on the training set. During the learning process, positive and negative detections are identified based on the IoU metric according to the equation given in Equation 3
p stands for class probability and y is for ground truth label.
x is for regression label
b is for bounding box, G is for ground truth, T+ and T− stand for positive and negative
threshold for IoU respectively.
Cascade R-CNN performs object detection by taking into account different IoU overlap levels compared to Faster R-CNN. Cascade R-CNN architecture proposes a series of customized regressors for different IoU overlap threshold values, as given in Equation 4. N represents the total number of cascades stages and d denotes for sample distribution in equation.
Dynamic R-CNN involves a dynamic training procedure compared to Faster and Cascade R-CNN. In this method, instead of using predefined IoU threshold values in the second stage where regressor and classifier are used, a dynamically changing procedure based on the distribution of region proposals is proposed given in Equation 3. For this purpose, The Dynamic Label Assignment (DLA) process given in Equation 5 is first used according to the T threshold value updated based on the statistical distribution of proposals for object detection. For localization task (bounding box regression), Dynamic SmoothL1 Loss given in in Equation 6 is used. Similarly to DLA, DSL also changes regression labels based on the statistical distribution of proposals.
Tnow is for current IoU threshold and updates according to data distribution
βnow represents the hyper-parameters to control smooth loss.
The main difference between one-stage object detection models and two-stage object detection models is that one-stage models do not involve a region proposal stage. Therefore, they generally provide faster detector performance compared to two-stage methods. One- stage object detection models show diversity in many aspects. These models can be divided into various sub-categories based on the differences in their architectures (such as Anchor Based, Anchor Free, Feature Pyramid, Context, etc.). The following is a summary of the one-stage object detectors used in this study.
The most important component of the RetinaNet architecture is a loss function called Focal Loss given in Equation 7. This proposed loss function solves the problem of class imbalance during training. It allows the rarer class to be learned better than the classic cross-entropy loss (Equation 1), leading to a balanced model training. The Focal Loss equation is defined as Focal Loss. RetinaNet is an anchor-based architecture that uses Resnet-based FPN as its backbone. While the backbone is used to extract convolutional features from the input, two different subnets that take the backbone’s outputs as inputs are used for object classification and bounding box regression.
YoloV3 is a variant of Yolo, which is an improved version of YoloV2 in various aspects. The most important factor that makes Yolo models faster than other object detection models is that they do not contain a complex pipeline. In this architecture, the entire image is used as the input to the network, and bounding box regression and object detection are directly performed on this image. The main differences between YoloV3 and YoloV2 are that YoloV3 uses a deeper architecture called Darknet-53 for feature extraction and predicts bounding boxes in 3 different scales. FCOS is an anchor-free object detection model that performs object classification and bounding box regression without requiring overlap (IoU) calculations with anchors or detailed hyper-parameter tuning. Anchor-based methods consider the center of anchor boxes as the location in the input image and use it to regress the target bounding box. However, FCOS considers any point inside the ground truth box as a location and uses the distance t∗ = (l∗, t∗, r∗, b∗) (given in Equation 8) for regression. If a point is inside multiple bounding boxes, it is considered an ambiguous example, and multi-level prediction is used to reduce this situation. FCOS also uses the centerness strategy to improve bounding box detection quality (given in Equation 9). To do this, FCOS adds a parallel stage to the classification task to predict whether a location is a center or not. The centerness value is trained using binary cross-entropy loss and added to FCOS’s loss function.
l∗, t∗, r∗, b∗ are distances calculated from bounding box.
The NAS-FPN model has made improvements on the FPN model architecture used in object detection models for extracting visual representations. Unlike other models that use manual FPN architecture design, this model narrows down a wide search space for FPN architecture through the Neural Architecture Search algorithm to extract effective FPN architectures.
The ATSS model proposes a method that performs adaptive sampling of training set examples by utilizing the statistical characteristics of positive and negative examples, based on the fact that they have a significant impact on the model’s performance. This approach has led to significant improvements in both anchor-based and anchor-free detectors, filling the performance gap between these two different architectures.
VarifocalNet proposed a different approach compared to other models in evaluating candidate objects during training by considering not only a classification score or a combination of classification and localization scores but also an IoU-based classification score (IACS) that takes detection performance into account. This approach takes both localization accuracy and object confidence score into consideration, resulting in successful results, especially in dense object detection scenarios. To estimate the IACS score, Varifocal loss (given in Equation 10) and star-shaped bounding box representation were proposed.
p represents to predicted IACS score and q stands for target score
Point 4: Ablation study is needed for verifying the proposed method.
Response 4: Thank you for your valuable feedback . In this study, four different ensemble strategies were proposed and their effectiveness was measured. If the effect of hyperparameters of the base models used in the ablation study is desired, as deemed appropriate by the reviewer, a total of 90 models were developed using 10-fold cross-validation for 9 architectures. With such intensive computational costs, hyperparameters were used as default settings as much as possible. Thus, the efficiency and use case of the ensemble strategies were demonstrated without intensive hyper parameter optimization. Nonetheless, training with default parameters produced promising results as stated in Table 4,5,6. For confidence score effects on moels, a new experimental setup was designed. This allowed for FROC curves to be presented for each individual anatomical and pathological region, containing information on FFPI vs TPR at various confidence scores, for the best individual models and ensemble strategies. The FROC curves were evaluated using the FAUC score (the area under the curve). Furthermore, experiments were conducted on the frequently used BraTS 2020 HGG dataset with 9 different models, and the ensemble strategy was compared to the best models by providing FROC graphs. The added text and graphs can be found below.
Figure 3. FROC curves based on TPR and FPPI for each anatomical and pathological region for the
best ensemble strategy and the best individual models on Gazi Brains 2020 datase
Point 5: The significance of this paper is not expounded sufficiently. The author needs to highlight this paper's innovative contributions
Response 5: Thank you for your valuable feedback. the introduction section has been extensively revised in order to better highlight the contribution of our study. I reviewed another 15 ensemble studies conducted on brain MRI datasets with regards to their objectives, methods, ensemble strategies, learning tasks, and time to publication. These studies were compared with the proposed study to better demonstrate its contribution. The contribution was presented more clearly by comparing it with relevant literature. . As a result, several changes have been made to the study. Article title has been changed to focus ensemble strategies for object detection. Thus, the ensemble strategies have been highlighted as the main focus of this work, and the relevant literature has been revisited to further support the problem definition.
Point 6: The authors should include some promising paper in the LR: Roy, Sudipta, and Samir Kumar Bandyopadhyay. "A new method of brain tissues segmentation from MRI with accuracy estimation." Procedia Computer Science 85 (2016): 362-369. // Roy, Sudipta, Debnath Bhattacharyya, Samir Kumar Bandyopadhyay, and Tai-Hoon Kim. "An iterative implementation of level set for precise segmentation of brain tissues and abnormality detection from MR images." IETE Journal of Research 63, no. 6 (2017): 769-783. // Roy, Sudipta, Tanushree Meena, and Se-Jung Lim. "Demystifying supervised learning in healthcare 4.0: A new reality of transforming diagnostic medicine." Diagnostics 12, no. 10 (2022): 2549.
Response 6: Thank you for reminding me of these valuable studies. I have cited them in the introduction section
Point 7: Please discuss the possible over-fitting risk by applying data growth study.
Response 7: Thanks for valuable feedback. To meause ensemble strategy on large dataset, experiments were conducted on the frequently used BraTS 2020 HGG dataset with 9 different models, and the ensemble strategy was compared to the best models by providing FROC graphs. This dataset contains much more slices compared to the Gazi Brains 2020 dataset but few labes. According to experimes as indicated in Fig. 4, the best model ensemble strategy is also effective for large dataset. In the discussion section, I have added text given in below.
As in many of the studies listed in Table 1, this study was also conducted with a single dataset. This can lead to over-fitting risks for deep learning models developed on limited datasets, limiting their generalization abilities. Although it may be possible to combine data from different open sources, finding datasets with similar labels or similar tasks is almost impossible. In this study, the performance of models was measured using the 10-fold cross-validation method on the Gazi Brains 2020 open dataset and BraTS 2020 HGG dataset. As expected, ensemble strategies are more effective in relatively small datasets like Gazi Brains 2020, but they also prove to be effective in larger datasets like BraTS 2020 HGG. This is particularly important in situations where data is limited, highlighting the necessity of ensemble strategies..
Figure 4. FROC curve for the pathological region for the best different model strategy on BraTS 2020
Dataset

Round 3
Reviewer 1 Report
The manuscript has been sufficiently improved to publication.
Reviewer 2 Report
I have no further comments